# Estimation of Natural Ventilation Rates in an Office Room with 145 mm-Diameter Circular Openings Using the Occupant-Generated Tracer-Gas Method

Hyeonji Seol [1,*], Daniel Arztmann [1], Naree Kim [2,3] and Alvaro Balderrama [1,4]

[1] MID Façade Design, Detmold School of Architecture and Interior Architecture, Technische Hochschule Ostwestfalen-Lippe, Emilienstraße 45, 32756 Detmold, Germany; daniel.arztmann@th-owl.de (D.A.); a.balderrama@tudelft.nl (A.B.)
[2] UBLO Inc., Seoul 03056, Republic of Korea; n.kim@ublo-window.com
[3] VS-A KOREA Ltd., Seoul 03056, Republic of Korea
[4] Architectural Façades and Products Research Group, Department of Architectural Engineering and Technology, Faculty of Architecture and the Built Environment, Delft University of Technology, Julianalaan 134, 2628 BL Delft, The Netherlands
[*] Correspondence: seolhyeonji@naver.com

**Abstract:** Natural ventilation in a building is an effective way to achieve acceptable indoor air quality. Ventilation dilutes contaminants such as bioeffluents generated by occupants, substances emitted from building materials, and the water vapor generated by occupants' activities. In a building that requires heating and cooling, adequate ventilation is crucial to minimize energy consumption while maintaining healthy indoor air quality. However, measuring the actual magnitude of the natural ventilation rate, including infiltration through the building envelope and airflow through the building openings, is not always feasible. Although international and national standards suggested the required ventilation rates to maintain acceptable indoor air quality in buildings, they did not offer action plans to achieve or evaluate those design ventilation rates in buildings in use. In this study, the occupant-generated carbon dioxide ($CO_2$) tracer gas decay method was applied to estimate the ventilation rates in an office room in Seoul, South Korea, from summer to winter. Using the method, real-time ventilation rates can be calculated by monitoring indoor and outdoor $CO_2$ concentrations without injecting a tracer gas. For natural ventilation in the test room, 145 mm-diameter circular openings on the fixed glass were used. As a result, first, the indoor $CO_2$ concentrations were used as an indicator to evaluate how much the indoor air quality deteriorated when all the windows were closed in an occupied office room compared to the international standards for indoor air quality. Moreover, we found out that the estimated ventilation rates varied depending on various environmental conditions, even with the same openings for natural ventilation. Considering the indoor and outdoor temperature differences and outdoor wind speeds as the main factors influencing the ventilation rates, we analyzed how they affected the ventilation rates in the different seasons of South Korea. When the wind speeds were calm, less than 2 m/s, the temperature difference played as a factor that influenced the estimated ventilation rates. On the other hand, when the temperature differences were low, less than 3 °C, the wind speed was the primary factor. This study raises awareness about the risk of poor indoor air quality in office rooms that could lead to health problems or unpleasant working environments. This study presents an example of estimating the ventilation rates in an existing building. By using the presented method, the ventilation rate in an existing building can be simply estimated while using the building as usual, and appropriate ventilation strategies for the building can be determined to maintain the desired indoor air quality.

**Keywords:** natural ventilation; occupant-generated $CO_2$ tracer gas method; ventilation rates; infiltration rates

## 1. Introduction

The air quality inside buildings where people spend much time is an essential determinant of healthy life and performance. Acceptable indoor air quality can be achieved by supplying outdoor air ventilation to dilute indoor contaminants to levels that are harmless to human health and do not negatively impact occupant senses of the indoor environment. Various contaminants can be generated inside buildings depending on the building's uses, materials, and conditions. Contaminants from the occupants, such as carbon dioxide, airborne viruses, and odor (bioeffluents), can deteriorate people's health and wellbeing [1,2]. Hazardous substances emitted from building materials or due to the operation of indoor appliances can lead to a broad range of health problems and may even be fatal. Furthermore, buildings need enough ventilation to avoid damage due to condensation on surfaces or in the structure that is liable to occur when the indoor air humidity is high.

Excessive carbon dioxide ($CO_2$) in the air inside a room due to a lack of ventilation may cause significant reductions in the decision-making performance of occupants, which may result in poor work performance [1]. More importantly, the indoor $CO_2$ concentration is an approximate surrogate for indoor pollutants from occupants and buildings [3]. A human generates pollutants such as carbon dioxide, airborne viruses, germs, and odor. Building materials and appliances can emit hazardous substances that may cause sick-house syndrome (SHS) and multiple chemical sensitivity (MCS) with various allergic irritations and diseases [4,5]. Kim-Jakyung and the Korean Ministry of Environment identified harmful substances emitted from building materials commonly used in Korea. Formaldehyde, volatile organic compounds (VOCs), toluene, and xylene styrene are emitted from finishing materials, insulation materials, and furniture. Radon and chromium are emitted from concrete and cement. In addition, hazardous substances are emitted from human activities, such as poisonous gases from heating and cooking. In addition, dust, mold, germs, bacteria, and odors can occur in aged buildings. Furthermore, excessive water vapor from indoor activities should be exhausted to avoid condensation.

However, the deterioration of indoor air is indiscernible and tends to be overlooked. On the other hand, we are strongly obligated to reduce energy consumption and heat loss against the climate crisis. Building technologies and regulations related to building envelopes in South Korea have been developed aiming to reduce heat loss by maximizing the airtightness of the envelope along with the global trends [6]. As the envelopes of buildings become more airtight, the indoor air is barely exchanged with the outside air unless occupants actively ventilate the indoor air. With such trends, the requirements for good indoor air quality and energy efficiency have often been considered in conflict with each other because ventilation could need more operation of HVAC equipment in cooled or heated buildings. Seppanen researched various strategies to reduce energy consumption while simultaneously improving indoor air quality in buildings. One of the strategies is to balance airflow, which is to ventilate the minimum airflow required for a room to prevent energy wasting caused by excessive ventilation [7].

In Korean regulations and guidelines, the required ventilation rates for existing or small-sized new buildings are not described nor enforced [8]. European and American standards recommend several methods for calculating required ventilation rates to achieve desired air quality in a building based on the number of occupants, building sizes, building uses, and building materials [9,10]. However, even if a required ventilation rate for a building is defined, designing or ventilating the building with a natural ventilation system accordingly to comply with the requirement is not easy. It is intricate to figure out how much air is exchanged between indoors and outdoors with natural ventilation. Moreover, the ventilation rates in a building vary depending on the condition of the building and the climate. Avella et al. measured and evaluated indoor air quality in Italian school buildings based on the indoor $CO_2$ concentration, air temperature, and relative humidity while recording the opening and closing status of windows in the classroom. The study calculated the theoretical ventilation rates that can be provided by opening the windows in a certain classroom at different temperatures, and it presented what percentage of openings

should be open to achieve acceptable indoor air quality based on the theoretical calculations. However, during the measurement period, neither the window opening was controlled nor the actual ventilation rates were calculated [11].

Two traditional techniques for measuring the infiltration and ventilation rates are fan pressurization tests and tracer gas methods. Fan pressurization tests are performed under the defined test conditions to measure the infiltration rate through the building envelope and are independent of climate variations [12,13]. The tests are performed at artificially-induced high pressures (commonly at 50 Pa and 75 Pa in a test zone) to overwhelm the natural pressure differences caused by wind and stack effects [14]. Fan pressurization test results may differ from actual in-service conditions. On the other hand, tracer-gas methods are carried out for both infiltration and ventilation rate measurements. These methods inject gas into the test zone and measure its concentration response [15,16]. The tracer-gas methods are conducted in actual environmental conditions and may better reflect seasonal variations in results [17]. Tracer gas methods can reflect the fluctuating infiltration and ventilation rates influenced by natural pressure-difference fluctuations, which is suitable for real-time evaluation in an actual room in service.

The occupant-generated $CO_2$ tracer gas method has received attention in that it is possible to estimate infiltration and ventilation rates reflecting the actual environmental conditions without injecting a tracking gas [18,19]. The method uses $CO_2$ generated by the occupants as a tracer gas, unlike the conventional tracer-gas methods that need injected tracer gas. It tracks the decay of the $CO_2$ concentration indoors after occupants leave the test zone [20]. The estimations can be done every hour in buildings during the unoccupied periods. However, the low concentration level of occupant-generated $CO_2$ and short-term estimation periods can lead to much uncertainty in calculations. Some studies reviewed the associated principles, assumptions, and uncertainties [21–25]. Valentina et al. calculated ventilation rates in Serbian schools with measured concentrations of $CO_2$ generated by occupants of the classrooms. However, the opening and closing status of the window was not recorded nor controlled. The varying ventilation rates due to weather change could not be known since the measurement was carried out for only five days. The uncertainties due to low concentration levels of $CO_2$ have not been considered [26].

The occupant-generated $CO_2$ tracer-gas method, which can be performed by anyone without many instruments, can help people to maintain desired IAQ while preventing excessive energy use. Occupants can make more accurate plans for natural ventilation by comparing the required ventilation rates and the estimated ones by measurements in their buildings. This paper aims to provide an example of indoor air quality and estimated ventilation rates in an office room with small circular openings in South Korea. Moreover, This paper compared infiltration and ventilation rates from summer to winter and evaluated the estimated ventilation rates compared with recommended ones by standards. The test room has eighteen small circular openings installed on the fixed insulated glasses for natural ventilation. The infiltration rates of the room were estimated while closing all the windows. In addition, the ventilation rates, including infiltration, were estimated while opening four circular openings on one side of the room. This study calculated the actual amount of ventilation under the predefined scenarios of window openings and analyzed the effect of varying environmental conditions on ventilation.

## 2. Materials and Methods

### 2.1. Climate, Site, Building, and the Test-Room Description

Airflow into a room through window openings and envelope leakage can vary depending on climate. The building of the test office room is located in the middle of Seoul, the largest city in South Korea. Seoul is classified as a temperate climate with four distinct seasons, with extreme temperature differences between summer and winter. Summer is hot and humid. Winter is cold and dry. The temperature goes up to as high as 35 °C in summer and goes down to as low as −20 °C in winter [27]. The wind-speed variation in Seoul is mild throughout the year. The windiest month of the year is February, with an

average hourly wind speed of 4.2 m/s. The calmest month of the year is June, with an average hourly wind speed of 3.1 m/s [28].

Together with the climate, indoor air quality is affected by outdoor air quality under natural ventilating conditions. The average air quality in South Korea in 2021 was evaluated as "moderate" as reported by IQAir, but the annual level of particulate matter, or PM 2.5, which is a main pollutant in the air of South Korea, exceeded the WHO PM2.5 guideline value by 3.8 times in 2021 [29].

The test room was built in 1982 with a reinforced-concrete structure. The four-story building is 15 m long along the north–south side, 9 m wide, and 13 m tall, with a total floor area of 505 m$^2$. The building has 150 mm of concrete wall clad with stone panels without insulation. The west and south facades of the building face a 20 m two-lane road and a 5 m road, respectively. The east and north facades are adjacent to other buildings.

Figure 1a shows the floor plan of the test room. The test room is on the 2nd floor of the building. The room's floor area is 84 m$^2$ with a 2.8 m height, and the volume is 235 m$^3$. The room is partly insulated internally with 50 mm of insulation. Columns and beams are not insulated, being expected to cause significant heat loss through them. The thermal-imaging camera showed that the inner surface temperature of the beam at the corner in the heated room on a winter day was close to 0 °C. The north, west, and south walls of the room face directly outdoors, and the other side of the east wall is a staircase of the building without air conditioning. The room has nine windows with fixed insulated glass. Seven large, fixed windows (2~2.5 m in width, 2 m in height) are on the west and south walls. Two long windows (2.3 m in width, 0.5 m in height) are on the north and east walls. Two small circular openings are on each fixed glass for natural ventilation. The glasses on the west and south walls have openings with different heights, and the glasses on the north wall have openings horizontally side by side. In other words, the room has ten small circular openings on the west wall, four small circular openings on the south wall, and two small circular openings on the north wall, as seen in Figure 1b. Figure 2 shows the elevation and section plans of the west wall. The diameter of the circular opening is 145 mm with an area of 0.017 m$^2$. The useful height for the stack effect between the upper opening and the lower one is 0.98 m.

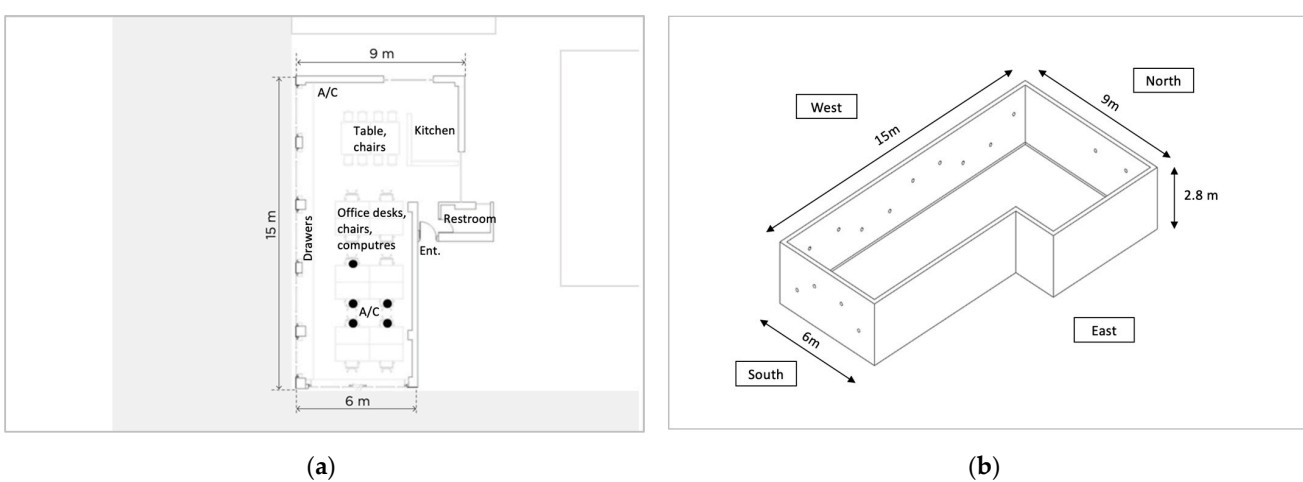

(**a**)  (**b**)

**Figure 1.** (**a**) Site plan and floor plan of the test room with the location of regular occupants; (**b**) Arrangement of small circular openings in the test room.

The test room is being used as an office. The room has an open working area, an unseparated kitchen, and a restroom. The room is furnished with desks, chairs, drawers, computers, a refrigerator, a sink, a TV, plants, and more. The concrete wall and furniture are painted. The floor is wood flooring. Five adult workers usually occupied the room during the measurement period. The regular workers usually sat in front of their desks, as shown in Figure 1a with circle symbols. Occupants occasionally moved around in the

room and went out for a while during work hours. Sometimes fewer people or more people were in the room. In November and December, we invited several people regularly to the office room on measurement days. The room was heated or cooled with two air conditioners, one against the north wall and one on the ceiling. Occupants used three fans in summer and four movable radiators in winter at their elbows. Occupants used the heating or cooling devices of their own accords over different seasons. The room does not have any mechanical ventilation system.

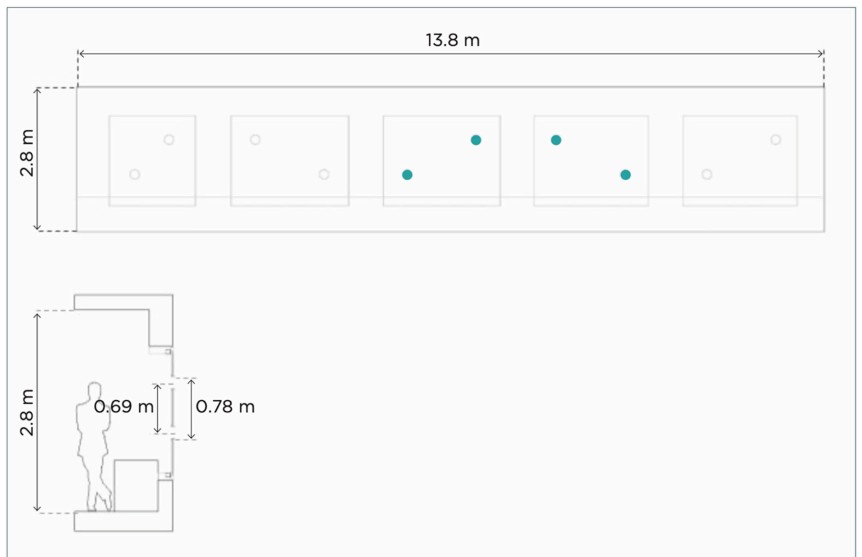

**Figure 2.** Elevation plan and section plan of the west wall in the test room. The openings opened for natural ventilation in this study are highlighted in blue.

*2.2. Require Ventilation Rates for the Test Room in Accordance with Regulations and Standards*

2.2.1. In Accordance with Korean Regulations

According to Standards for Facilities of Buildings [8], a new office building with a total floor area of over 3000 m$^2$ must equip with mechanical ventilation systems to ensure a ventilation rate of 29 m$^3$/person/h. In this case, a natural ventilation system (i.e., openable windows) cannot be substituted for a mechanical ventilation system. The standards also mandate specified ventilation rates for other buildings with more than defined floor areas besides office buildings. The mandated ventilation rates in new buildings in accordance with Korean regulations are summarized in Table 1. However, Korean regulations do not enforce the test room to satisfy any ventilation requirements due to its use and small size.

**Table 1.** Mandate ventilation rates in new buildings in accordance with Korean regulations (according to the Standards for Facilities of Buildings and School Health Act).

| Building Use and Size | Mandate Ventilation Rates | | |
|---|---|---|---|
| | L/s per Person | m$^3$/h per Person | ACH (m$^3$/h·V) |
| Office buildings over 3000 m$^2$ of floor area | 8.2 | 29 | |
| Exhibition halls or wedding halls over 2000 m$^2$ of floor area | 8.2 | 29 | |
| Supermarkets or shopping centers over 3000 m$^2$ of floor area | 8.2 | 29 | |
| Medical facilities over 2000 m$^2$ of floor area or with more than 100 sickbeds | 10 | 36 | |
| Schools | 6 | 21.6 | |
| Apartments over 50 households | | | 0.5 |

### 2.2.2. In Accordance with European Standard

European Standard EN 16789-1 [9] suggests three methods to maintain desired indoor air quality for nonresidential buildings. Design ventilation rates can be chosen depending on the desired satisfaction of the indoor air quality. Suppose the test room's occupants desire to maintain indoor air quality at the level of expected dissatisfaction below 15% (Category I). In that case, the standard recommends that the room's ventilation system should provide ventilation rates of 134 L/s (=482.4 m$^3$/h, 2.05 ACH) or 168 L/s (=604 m$^3$/h, 2.58 ACH). Those values were calculated by method 1 and method 3, respectively, provided that the floor area and volume of the room are 84 m$^2$ and 235 m$^3$ with five regular occupants. Alternatively, the occupants can keep $CO_2$ concentrations lower than 990 ppm to maintain the same level of indoor air quality (Category I), provided that the outdoor $CO_2$ concentration is 430 ppm, which is the mean outdoor $CO_2$ concentration during the measurement period. The minimum ventilation rates suggested to ensure the indoor air-quality level of expected dissatisfaction below 40% (category IV) are 37.7 L/s (=135.72 m$^3$/h, 0.58 ACH) and 46.2 L/s (155.52 m$^3$/h, 0.66 ACH), calculated by method 1 and method 3, respectively. Alternatively, the room's occupants can keep $CO_2$ concentrations lower than 1790 ppm to maintain the category IV indoor air quality. Design ventilation rates calculated by three methods for different target indoor air-quality levels are presented in Tables 2–4, respectively. Furthermore, the standard recommends ventilating even an unoccupied building room for diluting emissions from building materials. It recommends ventilating the test room at 12.6 L/s (45.36 m$^3$/h, 0.19 ACH) during unoccupied periods.

**Table 2.** Design ventilation rates for a five-person office with a floor area of 84 m$^2$ in a low-polluting building (nonadapted person) calculated by Method 1 in EN 16789-1.

| Category | Expected Percentage Dissatisfied | Design Ventilation Rate for Sedentary, Adults, Nonadapted Persons for Diluting Bioeffluents (a) | Design Ventilation Rate for Diluting Emissions from Low Polluting Building, LPB-2 (b) | Total Design Ventilation Rate for the Room = a × Number of Occupants + b × Floor Area | | |
|---|---|---|---|---|---|---|
| | % | L/s per Person | L/s·m$^2$ | L/s | m$^3$/h | ACH (m$^3$/h·V) |
| I | 15 | 10 | 1.0 | 134 | 482.4 | 2.05 |
| II | 20 | 7 | 0.7 | 93.8 | 337.68 | 1.44 |
| III | 30 | 4 | 0.4 | 53.6 | 192.96 | 0.82 |
| IV | 40 | 2 | 0.3 | 37.7 | 135.7 | 0.58 |

**Table 3.** Maximum indoor $CO_2$ concentrations calculated by Method 2 in EN 16789-1.

| Category | Expected Percentage Dissatisfied | Maximum $CO_2$ Concentrations in a Room Located in Seoul |
|---|---|---|
| | % | ppm |
| I | 15 | 990 [1] |
| II | 20 | 1240 [1] |
| III | 30 | 1790 [1] |
| IV | 40 | 1790 [1] |

[1] provided that the mean outdoor $CO_2$ concentration is 430 ppm.

**Table 4.** Predefined design ventilation rates for a five-person office with a floor area of 84 m$^2$ (nonadapted person) calculated by method 3 in EN 16789-1.

| Category | Expected Percentage Dissatisfied | Total Design Ventilation Airflow Rate per Person for an Office (a) | Total Design Ventilation Airflow Rate per Floor Area for an Office (b) | Total Design Ventilation Rate for the Room = Max (a × Number of Occupants; b × Floor Area | | |
|---|---|---|---|---|---|---|
| | % | L/s per Person | L/s m$^2$ | L/s | m$^3$/h | ACH (m$^3$/h·V) |
| I | 15 | 20 | 2 | 168 | 604.8 | 2.58 |
| II | 20 | 14 | 1.4 | 117.6 | 423.36 | 1.80 |
| III | 30 | 8 | 0.8 | 67.72 | 243.79 | 1.04 |
| IV | 40 | 5.5 | 0.55 | 46.2 | 155.52 | 0.66 |

### 2.2.3. In Accordance with American Standard

The American standard ASHRAE 62.1-2010 [10] also suggests three procedures to maintain acceptable indoor air quality for buildings. This standard only presented minimum ventilation rates instead of suggesting several values depending on desired satisfaction levels. If calculated according to the ventilation-rate procedure, the minimum design ventilation rate for the test room with five occupants is 37.7 L/s (=135.72 m$^3$/h, 0.58 ACH), as summarized in Table 5. Coincidentally, it is the same value for the minimum ventilation rate (category IV) by Standard EN method 1. In the IAQ procedure, ASHRAE listed the maximum air-contaminant concentrations that must not be exceeded, such as those of carbon dioxide, carbon monoxide, formaldehyde, lead, particles, and radon. Those values have been set by various national or international organizations concerned with outdoor and indoor air health and comfort effects. The maximum levels of carbon dioxide concentration from various organizations are presented in Table 6. Lastly, the natural ventilation procedure guides how to design a room to ensure minimum indoor air quality with natural ventilation. It proposes minimum distances from an occupied space to operable openings and minimum openable areas of window openings, which are different by the arrangement of the windows. Accordingly, occupied spaces in the test room should have openable windows closer than 14 m from the spaces, which is five times longer than the room's height. The openable area of windows must be at least 3.36 m$^2$, which is 4% of the net occupiable floor area.

**Table 5.** Ventilation rate required in the breathing zone of the occupiable space calculated by the Ventilation Rate Procedure in ASHRAE Standard 62.1.

| Occupancy Category | Minimum Ventilation Rates in Breathing Zone Depending on People | Minimum Ventilation Rates in Breathing Zone Depending on Floor Area | The Outdoor Airflow Required in the Breathing Zone = a × Number of Occupants + b × Floor Area | | |
|---|---|---|---|---|---|
| | L/s per Person | L/s m$^2$ | L/s | m$^3$/h | ACH (m$^3$/h·V) |
| Office buildings—office space | 2.5 | 0.3 | 37.7 | 135.72 | 0.58 |

**Table 6.** Comparison of limit values of $CO_2$ concentration in regulations and guidelines pertinent to indoor environments in Indoor Air Quality Procedure in ASHRAE Standard 62.1.

| Air Contaminants | Enforceable and/or Regulatory Levels | | | Nonenforced Guidelines and Reference Levels | | | |
|---|---|---|---|---|---|---|---|
| | NAAQS/EPA | OSHA | MAK | Canadian | WHO/Europe | NIOSH | ACGIH |
| Carbon dioxide | - | 5000 ppm | 5000 ppm | 3500 ppm | - | 5000 ppm | 5000 ppm |

*2.3. Theory*

Carbon dioxide ($CO_2$) is released by human breathing. The $CO_2$ concentrations in an occupied room may become higher than outside when the room is unventilated or not adequately ventilated. Then when occupants leave the room (i.e., when the source of $CO_2$ disappears), the $CO_2$ concentration decreases as the indoor air is exchanged with the outdoor air through leaks in the building envelope and opened windows. Here, the decay rates of the $CO_2$ concentration during unoccupied periods can be used to calculate the infiltration and ventilation rates.

According to Park et al.'s study [21], air change per hour (ACH) can be estimated by measuring the decreasing indoor $CO_2$ concentrations after occupants leave the room and the outdoor $CO_2$ concentrations at that time with Equation (1).

$$\text{ACH} = -\frac{d(C_{in} - \overline{C}_{out})}{dt} \cdot (1 + x) \tag{1}$$

Here, $C_{in}$, $C_{out}$, and t are indoor $CO_2$ concentration, outdoor $CO_2$ concentration, and time, respectively. The x term corrects the estimation error that arises when the time varying outdoor $CO_2$ concentration is assumed constant. The x term can be considered as the mean value for the estimation periods in Equation (2):

$$x = \frac{1}{N} \sum \frac{C_{out} - \overline{C}_{out}}{C_{in} - \overline{C}_{out}} \tag{2}$$

Here, N is the number of the collected measurements during the estimation time range. If the mean of the outdoor $CO_2$ concentration during the estimation time range is used for the mean $\overline{C}_{out}$, the x term vanishes. In this study, the x term in Equation (2) was not considered because the daily mean of the outdoor $CO_2$ concentration was used for the mean $\overline{C}_{out}$, assuming outdoor $CO_2$ concentration is the same over a day. We measured outdoor $CO_2$ concentrations near the test room on 31 July 2022. $CO_2$ concentrations were constant within measurement-error ranges throughout the day. It may be because of little planting in the vicinity of the test room or little other factors that could cause changes in the concentration of $CO_2$ during days and nights. Ultimately, this paper calculated ventilation rates (in ACH) using Equation (3):

$$\text{ACH} = -\frac{d(C_{in} - \overline{C}_{out})}{dt} \tag{3}$$

*2.4. Measurement Setup and Calculations*

We assumed that the distribution of $CO_2$ was constant everywhere in the test room. We turned on three fans vigorously before leaving to achieve uniform $CO_2$ concentrations in the test room. Infiltration and ventilation rates were estimated only using the indoor $CO_2$ concentrations data above 1000ppm, as recommended in Park et al.'s study [21], to prevent underestimation due to the uncertainty of measurement data under extremely low concentration conditions. Whether an estimate is reliable can be judged by the linearity of the regression equation $\ln(C_{in} - \overline{C}_{out})$ during the time range for the estimate. The estimates that satisfy $R^2 \geq 0.99$ of $\ln(C_{in} - \overline{C}_{out})$ were selected for quantitative analysis, as recommended in Park et al.'s study [21].

We conducted indoor measurements in the test room for twenty-six days from August to December 2022 and estimated the infiltration and ventilation rates of the room. Infiltration rates of the room were estimated for eight days with all the windows closed, which is the opening scenario 1, as shown in Figure 3a,b. Ventilation rates, including infiltration, were estimated for eighteen days, with four circular openings on the west wall open, which is the opening scenario 2, as shown in Figure 3c,d. The openings opened for one-sided natural ventilation are highlighted in blue in Figure 2.

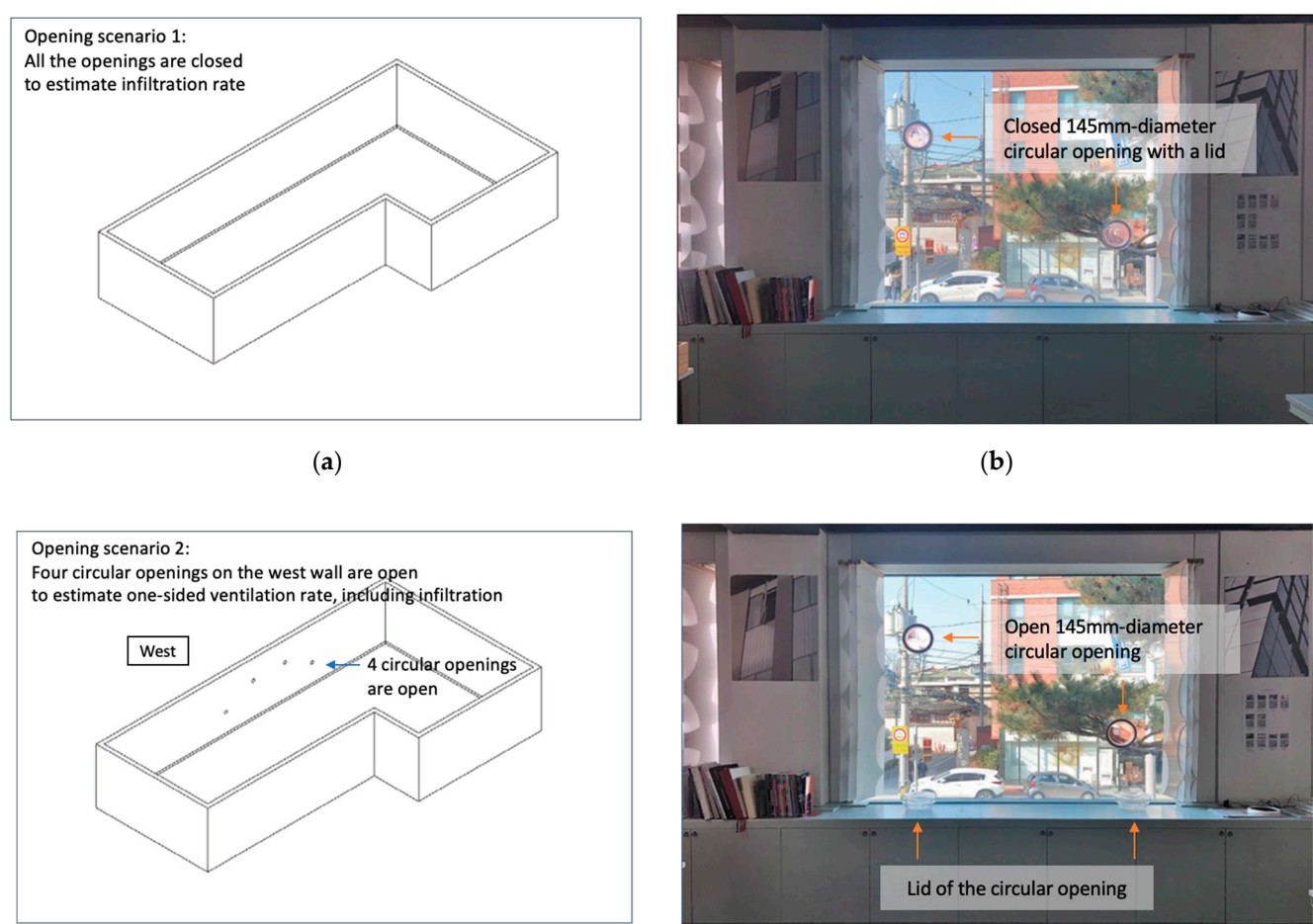

**Figure 3.** (**a**) Opening scenario 1: when all the windows are closed to estimate infiltration rate; (**b**) Closed circular openings; (**c**) Opening scenario 2: when four circular openings are open to estimate ventilation rate, including infiltration (**d**) Opened circular openings.

We installed four air-quality monitors (U1~U4) onboard multiple sensors to measure indoor temperature, humidity, $CO_2$ concentration, and PM concentration. The four monitors were installed in different locations in the test room with the same height of 0.7 m, as seen in Figure 4. First, the onboard sensor for measuring temperature and humidity is a BME 280 from Bosch, located in Gerlingen, Germany, with a −40~85 °C, 0~100% measuring range and a ±1 °C, ±3% measurement accuracy, respectively. We calibrated the BME 280 sensors in the Korean Laboratory Accreditation Scheme, which signed the ILAC-MRA. Another onboard sensor for measuring $CO_2$ concentration is SCD30 from Sensirion, Stäfa, Switzerland, with a 400~10,000 ppm measuring range and ±30 ppm measurement accuracy. We calibrated the SCD30 sensors with the $CO_2$ values measured at Namsan Tower, which is 3.3 km away from the test building. We measured $CO_2$ concentrations in the well-ventilated test room on 31 July 2022 and calibrated the sensors to have the same daily mean $CO_2$ concentration as the one measured at the Namsan Tower on the same day. We assumed the indoor $CO_2$ concentrations on 31 July 2022 were the same as the outdoor ones because the test room was unoccupied, and all the windows had been open from two days ago. Last, the onboard sensor for measuring PM concentration is SPS30 from Sensirion, located in Stäfa, Switzerland, with a 0~1000 μg/m³ measuring range and ±10% measurement accuracy.

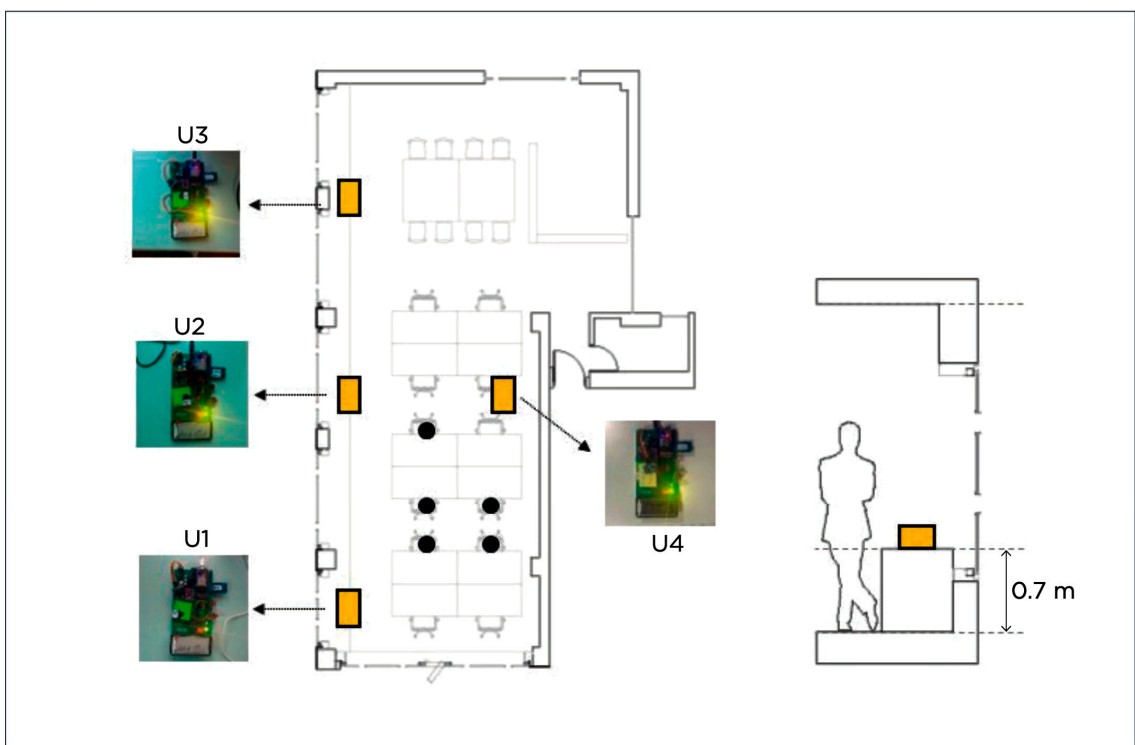

**Figure 4.** Location of air-quality monitors U1–U4 and occupants in the test room.

On the other hand, outdoor temperature, wind speed, and PM concentration were measured every minute at the outdoor meteorological station 2 km away from the test building. The Korean meteorological administration uploads the measurement data on its website [30]. The outdoor $CO_2$ concentration was measured at the Namsan Tower, 3.3 km away and 380 m higher than the test room. The Climate Lab of Seoul University posts the daily mean of measured $CO_2$ concentrations on its website [31].

Among measurements during the measurement periods, we collected the measurements only during unoccupied periods and when indoor $CO_2$ concentrations were more than 1000 ppm to estimate infiltration and ventilation rate. We will call the period when unoccupied and indoor $CO_2$ concentrations were more than 1000 ppm as an estimation period in this study. Since occupants usually left the test room between 6 to 8 pm, the estimation period was in the early evening on most measurement days. An estimate was calculated with measurements every 10 min for an hour. In other words, seven measurements with 10-min intervals were used for calculating an estimate. Park et al. quantitatively analyzed that the estimates calculated with a 1-h time range have lower than 20% uncertainty [5]. The 1-h time ranges overlapped with the 10-min intervals within the estimation period. Figure 5 illustrates the terms about periods and intervals of measurement and estimation with a diagram.

The timed datasets (i.e., seven measurements for a 1-h-time range) were 340 sets and it was for 77 h and 40 min, excluding overlapped time. In this study, $R^2 \geq 0.99$ of the regression equation $\ln(C_{in} - \overline{C}_{out})$ was set as the condition to satisfy the linearity. The value of $R^2$ was rounded to two decimal places. Among 340 estimates from the timed datasets of the time ranges, 103 estimates did not satisfy the linearity and were eliminated for the quantitative analysis.

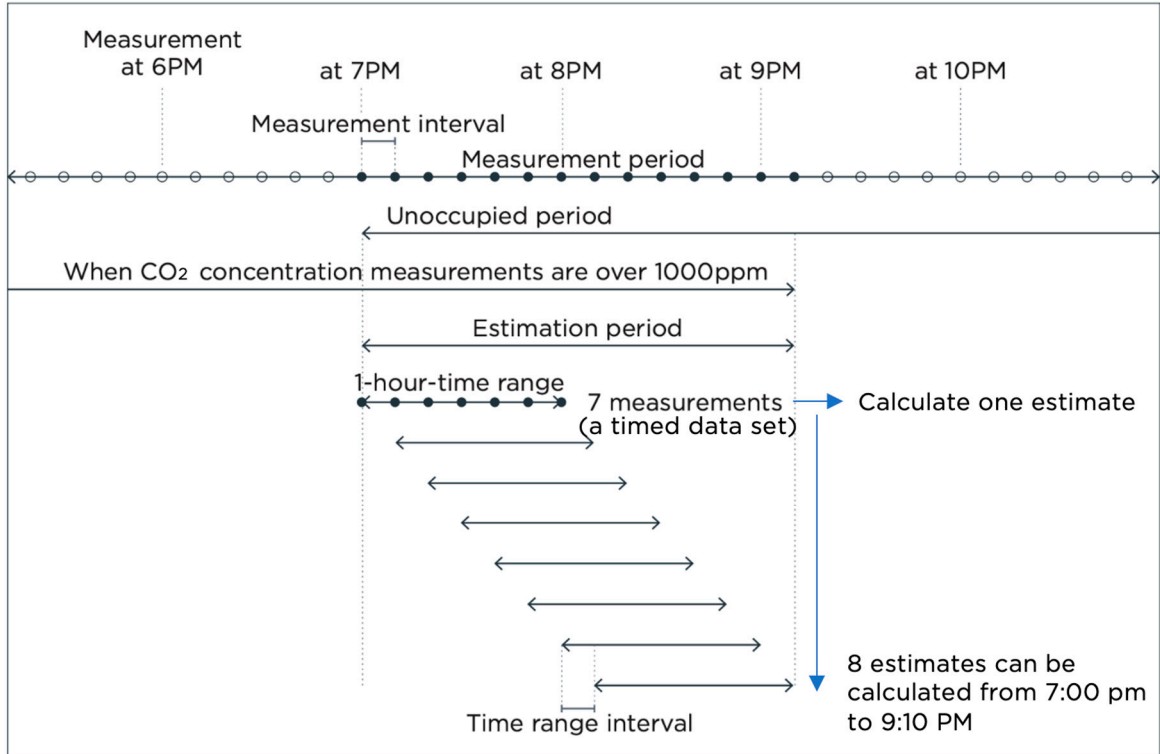

**Figure 5.** Terms about periods and intervals of measurement and estimation.

## 3. Results

The measurements and estimations were conducted for twenty-six days from August to December 2022, with seven days in August, six days in October, nine days in November, and four days in December. The dates, the opening scenarios, decreases in $CO_2$ concentrations during unoccupied periods, and mean outdoor $CO_2$ concentrations on each day are listed in Appendix A. The highest indoor $CO_2$ concentration measured in the test room was 2673 ppm during an occupied period and 2592 ppm during an unoccupied period. The mean outdoor $CO_2$ concentration from August to December measured at Namsan Tower was 430 ppm, with 428 ppm in August and 442 ppm in December.

The date, indoor and outdoor environmental conditions, estimation periods, and the summary of estimates are listed in Appendix B. During the measurement days, the mean outdoor temperature of measurement days was 13.3 °C, with 28.4 °C on the hottest day and −4.6 °C on the coldest day. The highest indoor temperature was 33.4 °C in the afternoon of August and the lowest one was 7.7 °C in the morning of December. The mean outdoor windspeed of measurement days was 2.1 m/s, with 3.5 m/s on the windiest day, and 1.3 m/s on the calmest day.

The highest daily mean of estimated infiltration rates was 0.21 ACH on December 15 and the lowest one was 0.07 ACH on August 4. The highest daily mean of estimated ventilation rates, including infiltration, was 0.61 ACH on December 13 and the lowest one was 0.22 ACH on August 14.

### 3.1. Indoor Air Quality during an Occupied Period in the Test Room

Figure 6 shows various indoor air qualities in the test room and outdoor weather conditions from the morning to evening on 11 August 2022. It was one of the hot and humid summer days in South Korea. Five workers occupied the room from 9 am to 7 pm. Workers turned on cooling devices as they were comfortable during their work hours. The indoor air-quality indicators in the chart were measured by sensor U2 located in the middle of the room. We controlled ventilating conditions from 2 pm.

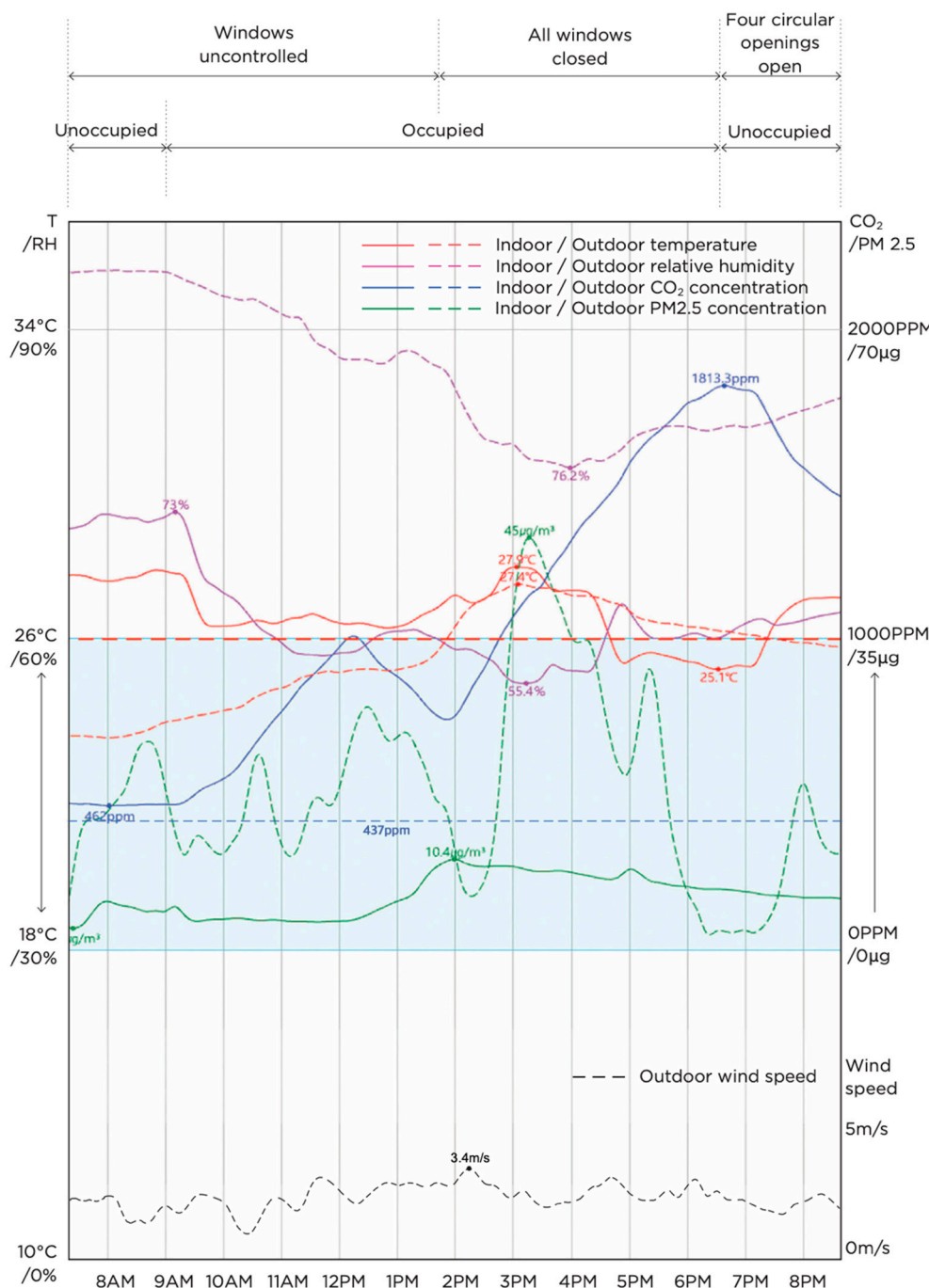

**Figure 6.** Indoor air quality and outdoor weather conditions during an occupied period in the test room on 11 August 2022.

The red, purple, blue, green, and black lines show the every-minute measurements of temperature, relative humidity, $CO_2$ concentration, PM 2.5 concentration, and wind speed, respectively. Dotted lines in each color represent outdoor measurements, and solid lines represent indoor measurements. The highest and lowest values of the day are written above and below the dots on the lines. The area highlighted in light blue represents the comfort zone, a range of comfortable indoor environments based on Korean regulations and guidelines [32,33], where the temperature is between 18 °C and 26 °C, relative humidity is between 30% and 60%, $CO_2$ and PM 2.5 concentration are less than 1000 ppm and 35 μg/m$^3$, respectively.

During the occupied period on 11 August 2022, the indoor temperature was slightly higher than the comfort zone most of the time, even though air conditioners were being operated to cool the room. The indoor relative humidity remained high because of the humid weather. The indoor and outdoor temperature and wind speed fluctuated throughout the measurement period.

Indoor $CO_2$ concentrations were significantly affected by occupancy and ventilating conditions. Before 9 am, when no one was in the room, the $CO_2$ concentration was close to the outdoor one. Then, as workers started to enter the room from 9 am, it started to increase. Later, it decreased again during lunchtime as workers went out severally and opened the door frequently between 12 pm and 2 pm. As all workers returned to the room after 2 pm, we closed all the windows, and the $CO_2$ concentration continued to increase until the workers left the room.

The indoor $CO_2$ concentrations trend implies that even if only five people were in such a spacious room, indoor air quality could deteriorate significantly when all the windows were closed. At noon, the $CO_2$ concentration already reached 990 ppm, which is the maximum $CO_2$ value for category I indoor air quality based on Standard EN (see Section 2.2). The $CO_2$ concentration at 7:30 pm reached 1813 ppm, higher than the maximum $CO_2$ value for the minimum indoor air quality (category IV) based on standard EN. After all, at 7 pm, when all the occupants left the room, $CO_2$ concentrations started to decrease. We left the four circular openings open for natural ventilation when we left the room.

PM 2.5 concentrations were usually lower inside than outside because of the higher PM concentrations outside. During lunchtime, between 1 pm and 2 pm, when outside air entered the room due to people coming in and out, the indoor PM 2.5 concentration rose, then decreased slowly when the windows were closed after 2 pm. Still, the outdoor PM concentration was acceptable on that day. Thus, it would have been much better to open the windows for natural ventilation to dilute bioeffluents and other possible pollutants from buildings.

### 3.2. Decays of Indoor Carbon Dioxide Concentration during Unoccupied Periods

We closed all the windows in the test room during the daytime to increase the $CO_2$ concentration during the measurement period. Then, when all the workers left the room, we left the room with all the windows closed or four circular openings open to measure the $CO_2$ decays under the two different opening scenarios during unoccupied periods. For example, Figure 7 shows the improving indoor air quality during an unoccupied period from the evening on 11 August 2022 when we left the room with four circular openings open. The solid blue line shows the decay of the indoor $CO_2$ concentration. The indoor $CO_2$ concentration peaked around 7 pm, just before the occupants left the room, and decreased until it became close to the outdoor $CO_2$ concentration in the form of a logarithmic graph. The initial $CO_2$ concentration just after all the occupants left the office peaked at 1811 ppm at 7 pm. And then, it decreased to 1569 ppm after one hour, 1435 ppm after two hours, and 1293 ppm after three hours. This way, we collected the $CO_2$ concentration decay data every measurement day.

Figures 8 and 9 show the decays of $CO_2$ concentrations during unoccupied periods of twenty-six measurement days. Figure 8 shows the $CO_2$ concentration decay under the opening scenario 1, when all the windows were closed, whereas Figure 9 shows the decay under the opening scenario 2, when four circular openings on one side were open. Circle symbols represent initial $CO_2$ concentrations when all the occupants just left the room. Triangles, upside-down triangles, and squares represent the decreased $CO_2$ concentrations after one hour, two hours, and three hours, respectively. More measurement values are summarized in a table in Appendix A.

In August and October, when 3–5 regular workers were in the room, the initial $CO_2$ concentration increased to 1200~1800 ppm. On some days in November and December, when many people visited the room, the initial $CO_2$ concentration exceeded 2000 ppm. The average 3-h decrease during unoccupied periods when all the windows were closed

was 366 ppm, and the decrease when four circular openings were open was 800 ppm. The greatest three-hour decrease was 1876 ppm on 13 December 2022, when the four circular openings were open. It was a windy and cold day, and the initial $CO_2$ concentration was the highest among the measurement days. Nevertheless, a greater decrease in $CO_2$ concentration does not always mean a greater magnitude of ventilation. Even if the same amount of ventilation occurs, if the indoor $CO_2$ concentration is much higher than the outdoor one, the indoor one decreases more quickly.

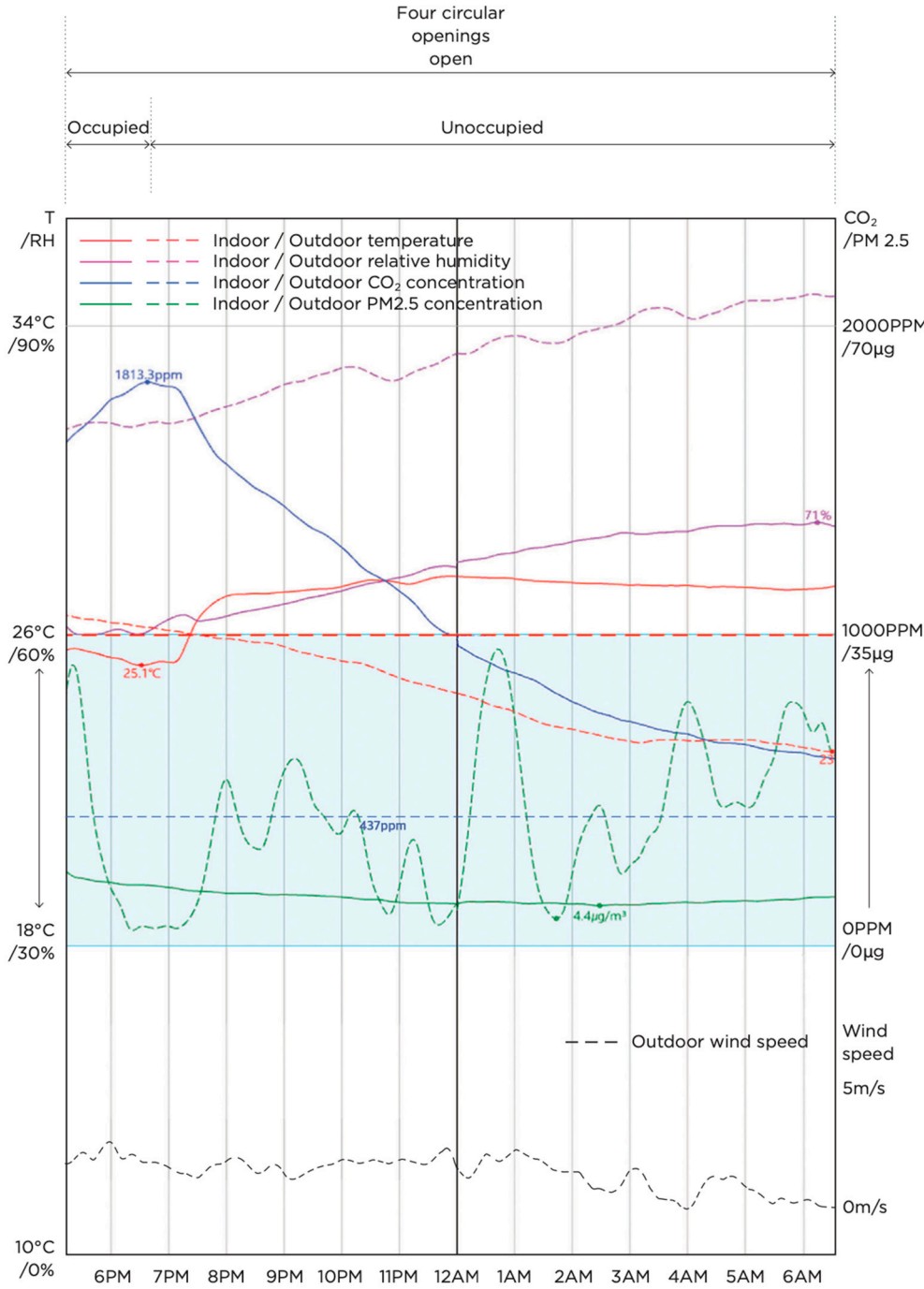

**Figure 7.** Decay of indoor $CO_2$ concentration and indoor and outdoor air qualities during an unoccupied period in the test room on 11 August 2022.

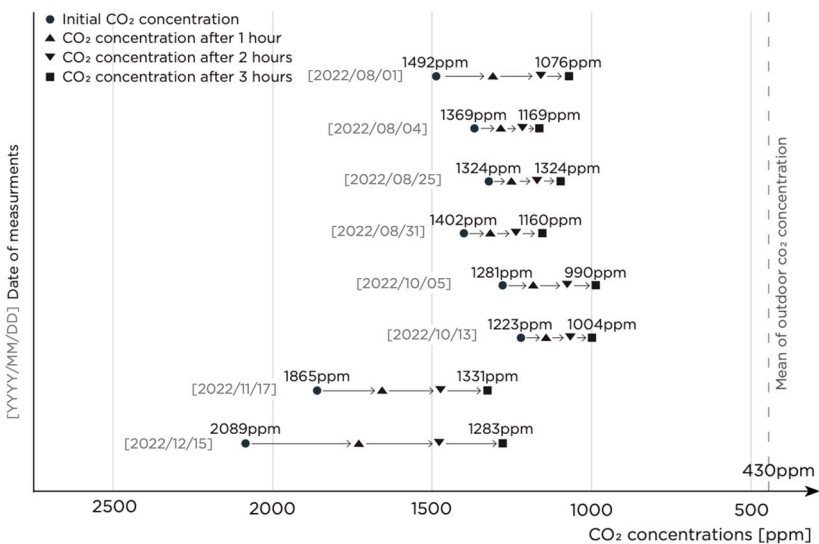

**Figure 8.** Decays of indoor $CO_2$ concentration during unoccupied periods when all the windows were closed (opening scenario 1).

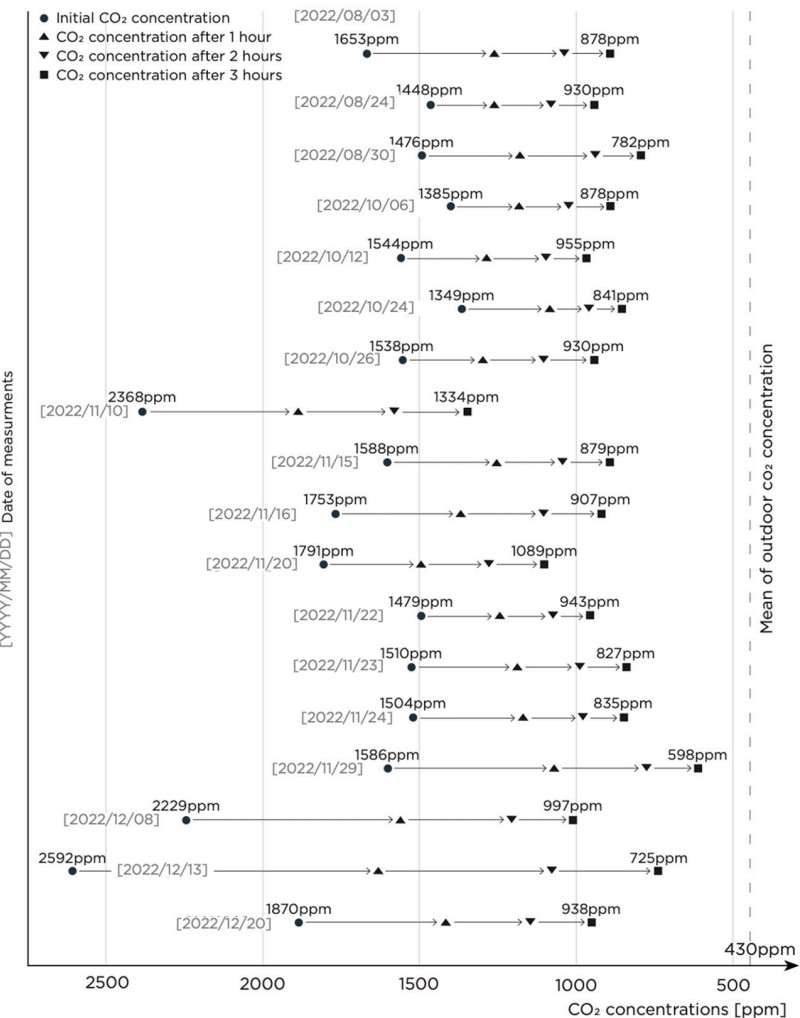

**Figure 9.** Decays of indoor $CO_2$ concentration during unoccupied periods when four circular openings were open (opening scenario 2).

### 3.3. Estimated Infiltration Rates When All the Windows Were Closed (Opening Scenario 1)

One hundred twenty-nine valid infiltration-rate estimates were calculated using the occupant-generated $CO_2$ tracer-gas decay method during the measurement period. The estimates are coordinated as dots in scattered charts, as shown in Figures 10 and 11. The charts show the correlation between estimated infiltration rates and the environmental conditions (temperature difference between indoor and outdoor, and outdoor wind speed). The x-axis of Figure 10 shows the temperature difference between inside and outside the test room, and the x-axis of Figure 11 shows the outdoor wind speed. The y-axis shows the estimated infiltration rates in two different units. Additionally, the dots of estimates are expressed in different shades to express the other condition that is also changing simultaneously. The darker dots represent higher wind speeds in Figure 10 and the greater temperature difference in Figure 11.

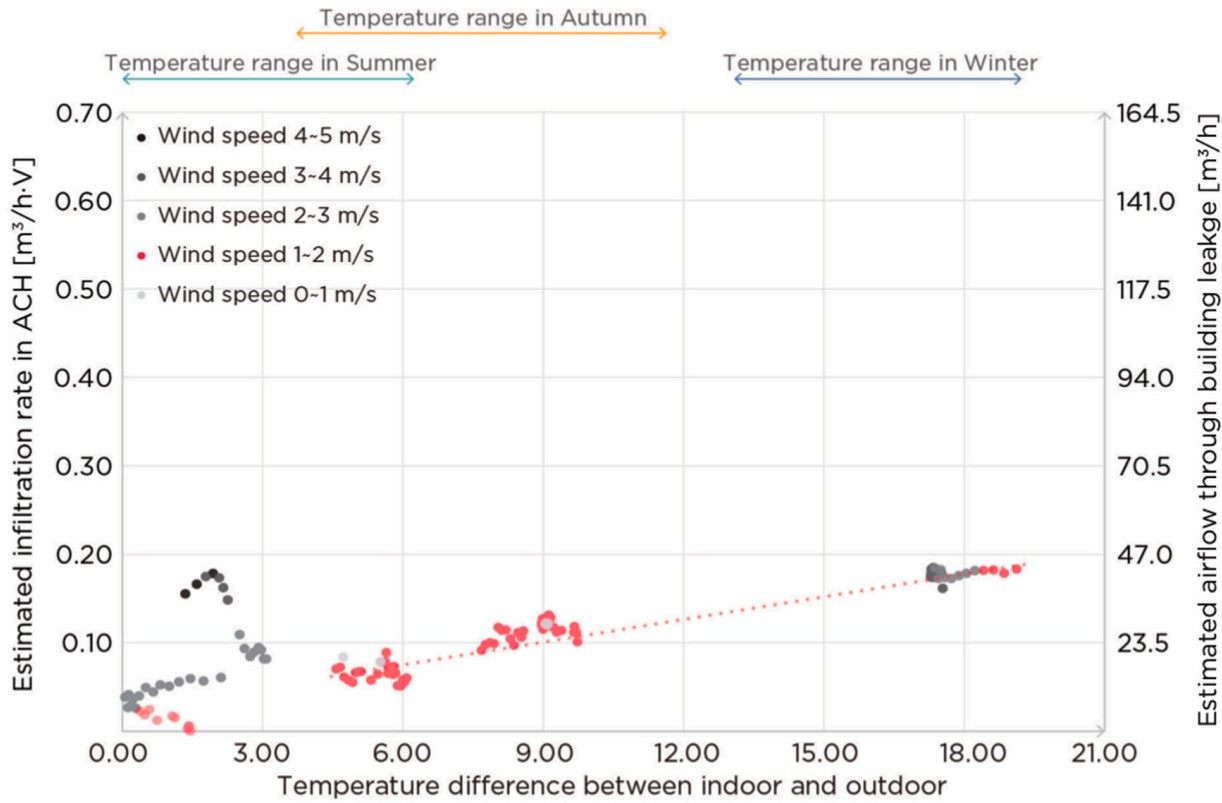

**Figure 10.** The effect of temperature differences on estimated infiltration rates.

In Figure 10, when the wind speeds were less than 2 m/s (colored dots in red) and the temperature differences were greater than 3 °C, the infiltration-rate estimates showed an increasing trend as the temperature differences increased. Meanwhile, when the temperature differences were less than 3 °C, it was difficult to trace a relationship between the temperature difference and estimates.

In Figure 11, when the temperature differences were less than 3 °C (colored dots in red), and the wind speeds were greater than 1.5 m/s, the infiltration-rate estimates showed an increasing trend with increasing wind speeds. Meanwhile, when the wind speeds were less than 2 m/s, it was difficult to trace a relationship between wind speed and estimates. Rather, dots of different shades were vertically layered, implying that the estimates were more affected by the temperature differences when the outdoor wind speeds were calm.

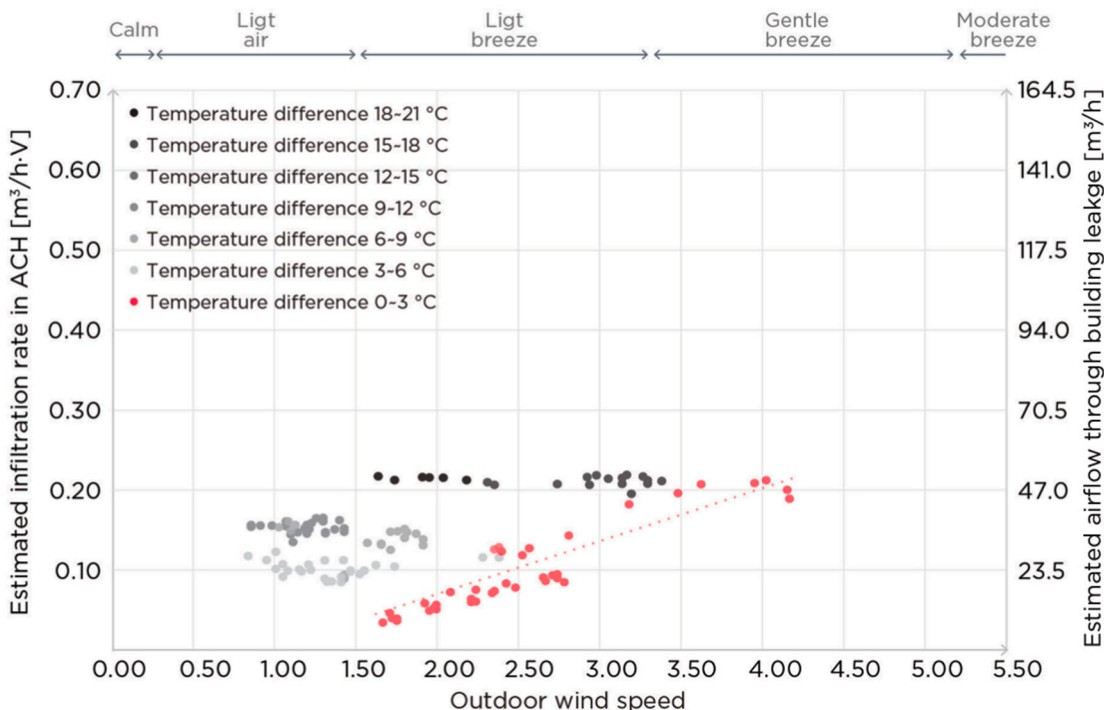

**Figure 11.** The effect of wind speeds on estimated infiltration rates.

### 3.4. Estimated Ventilation Rates, including Infiltration, When Four Circular Openings on One Side Were Open (Opening Scenario 2)

One hundred eight valid ventilation-rate estimates, including infiltration, were calculated using the occupant-generated $CO_2$ tracer gas decay method during the measurement period. The estimates are coordinated as dots in scattered charts, as shown in Figures 12 and 13. The charts show the correlation between estimated ventilation rates and the environmental conditions. The x-axis of Figure 12 shows the temperature difference between inside and outside the test room, and the x-axis of Figure 13 shows the outdoor wind speed. The y-axis shows the estimated infiltration rates in two different units. Additionally, the dots of estimates are expressed in different shades to express the other condition that is also changing simultaneously. The darker dots represent higher wind speeds in Figure 12 and the greater temperature differences in Figure 13.

In Figure 12, when the wind speeds were less than 2 m/s (colored dots in red and green) and the temperature differences were greater than 6 °C, the ventilation-rate estimates showed an increasing trend as the temperature differences increased. When the wind speeds were more than 2 m/s, it was difficult to find a relationship between the temperature difference and the ventilation rate based on the estimates obtained in this study.

In Figure 13, when the temperature differences were less than 6 °C (colored dots in red) and the wind speeds were greater than 1.5 m/s, the ventilation-rate estimates showed an increasing trend as the wind speeds increased. When the wind speeds were less than 1.5 m/s or when the temperature differences were more than 6 °C, it was difficult to find a relationship between wind speed and ventilation with the estimates obtained in this study.

Estimated ventilation rates more than doubled on days when wind speeds were higher than 4 m/s and the temperature differences were more than 18 °C. These estimates were calculated with winter measurements, implying that even with the same openings, ventilation rates could more than double depending on the weather.

Moreover, ventilation-rate estimates were more scattered than infiltration-rate estimates, and it was difficult to trace a trend. It implies that temperature difference and wind speed were the main factors that significantly affected infiltration rates. However, other factors may have influenced ventilation rates more significantly, such as the direction of

the outdoor wind, the airflow inside the room formed by ventilation, and the uneven air and temperature distribution inside the room.

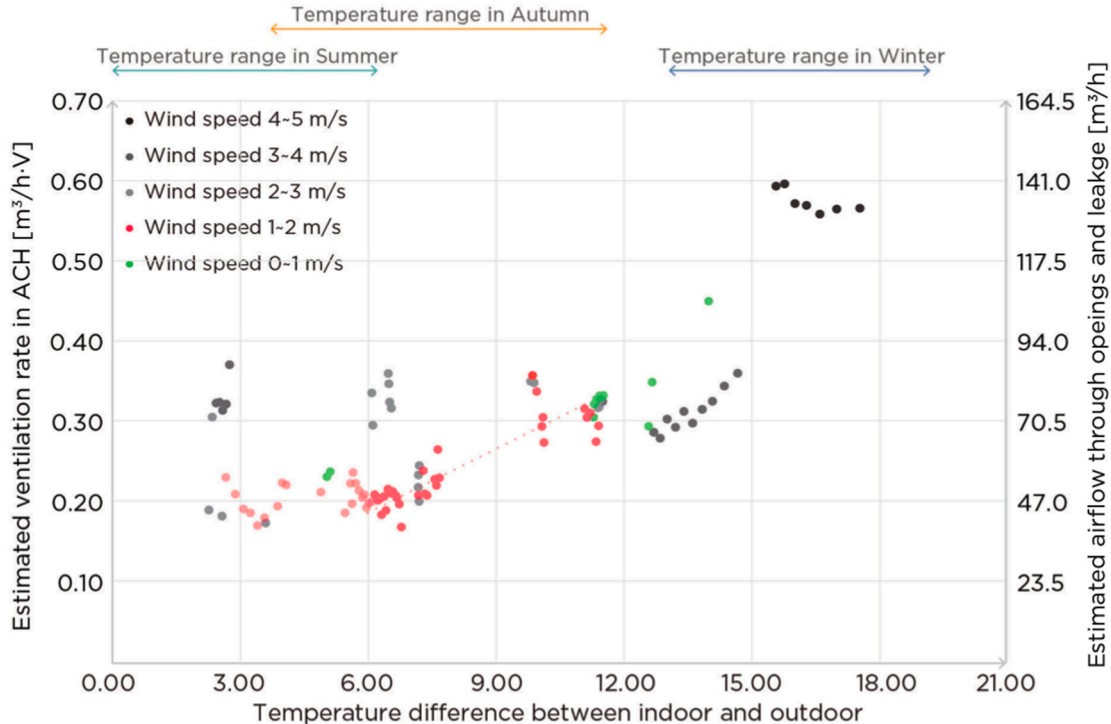

**Figure 12.** The effect of temperature differences on estimated ventilation rates, including infiltration, when four circular openings on one side were open.

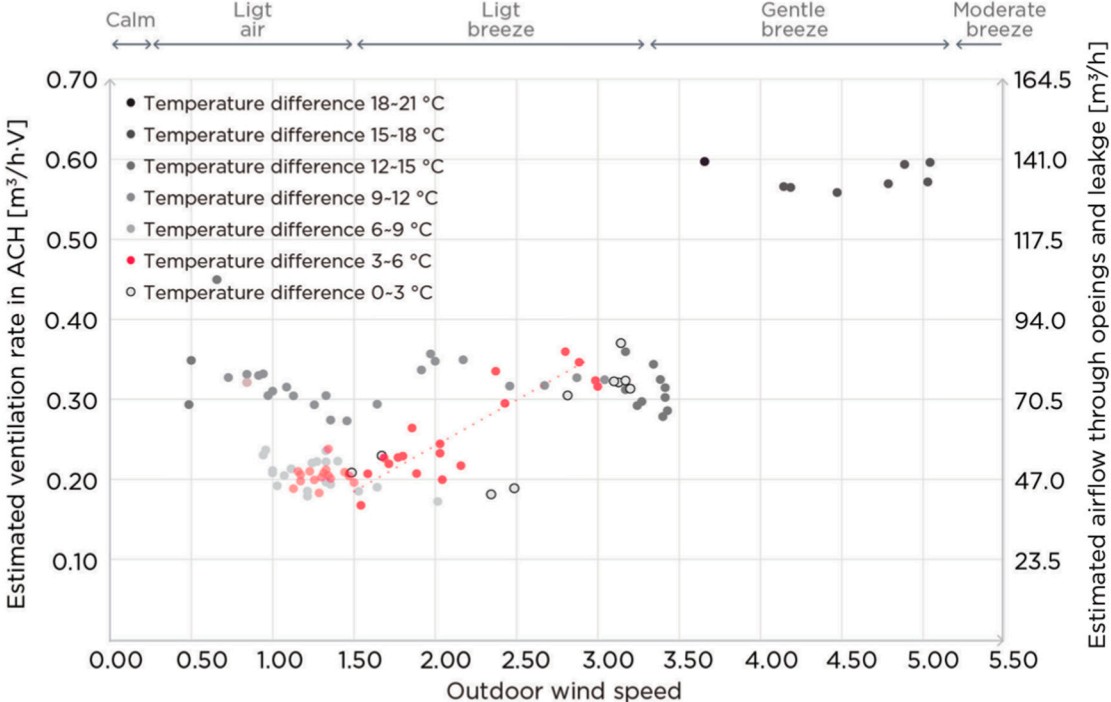

**Figure 13.** The effect of wind speeds on estimated ventilation rates, including infiltration, when four circular openings on one side were open.

### 3.5. Various Infiltration and Ventilation Rates in Different Seasons of South Korea

Figure 14 shows various environmental conditions in different seasons of Republic of Korea of the test room and, subsequently, various infiltration and ventilation rates estimated in each season. The size of a bubble represents the magnitude of mean-estimated infiltration or ventilation rates during one measurement day; so twenty-six bubbles of each day were coordinated according to their environmental conditions during their corresponding estimation periods. The values beside and inside the bubbles are the average of valid infiltration and ventilation estimates during a day, respectively. Green, orange, and blue bubbles represent summer, autumn, and winter estimates, respectively. Pale bubbles represent infiltration rates, and dark bubbles represent ventilation rates.

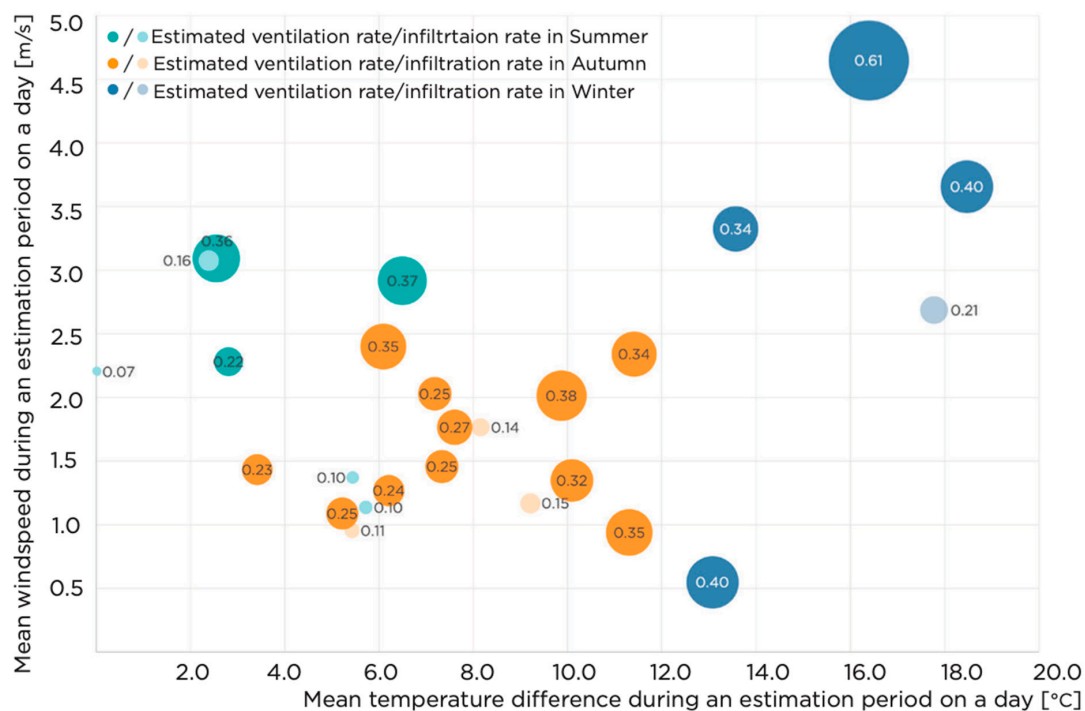

**Figure 14.** Daily mean of estimated infiltration and ventilation rates in different seasons.

As shown in Figure 14, green bubbles are on the left side, orange bubbles are in the middle, and blue bubbles are on the right side of the chart, indicating that the test room's temperature difference increased from summer to winter. More green and blue bubbles are in the upper part of the chart than orange bubbles, indicating that the summer and winter were windier than the autumn. More importantly, the sizes of the bubbles increase as they go up and to the right, indicating that the infiltration or ventilation rates increased as the temperature difference and wind speed increased. The pale bubbles are generally smaller than darker ones, indicating that the infiltration rates were less than the ventilation rates. For reference, the estimated ventilation rates include leakage of the room.

The temperature differences in the summer among the measurement days were slight, with an average of 3.6 °C (0~6.5 °C), while the wind speed averaged 2.3 m/s (1.1~3.1 m/s). The result in Section 3.4 showed when the temperature differences were less than 6 °C and the wind speeds were greater than 1.5 m/s, the ventilation-rate estimates showed an increasing trend as the wind speeds increased. Based on the result, it can be assumed that the estimated ventilation rates in the summer generally increased with a more significant influence on wind speeds. The bubbles which get bigger as they go up in the chart confirm that assumption. On the other hand, in autumn, the temperature differences increased to an average of 7.8 °C (3.4~11.4 °C), while the wind speeds were calmer on most days, being averaged 1.6 m/s (0~2.4 m/s). It can be assumed that the primary factors that would have significantly affected the ventilation rates would have changed depending on the day's

weather. At any rate, the estimated ventilation rates in summer and autumn were similar because the two factors complemented one another.

The temperature differences in winter increased significantly to an average of 15.9 °C (13.1~18.5 °C). On top of that, the wind speeds increased to an average of 3m/s (0.6~4.7 m/s) with a wide variance. As a result, the estimated ventilation rates in winter increased significantly compared to summer and autumn, even though the same openings were open. In addition, the variance of the magnitude of estimates, or bubble sizes, increased due to the wide variation in wind speeds. Yet, it is hard to assume which factor had a greater effect on winter ventilation rates only with this study's results.

## 4. Discussion

*Evaluation of the Estimated Ventilation Rates in the Test Room in Accordance with Standards EN and ASHRAE*

The minimum ventilation rate suggested in standard EN and ASHRAE is 0.58 ACH. Among the eighteen days of estimating the ventilation rates with the opening scenario 2 (the four circular openings open), the daily mean ventilation rate on 13 December only exceeded the minimum requirement, with 0.61 ACH. Figure 15 compares the mean estimated infiltration and ventilation rate of different seasons. The mean estimated infiltration rates of summer, autumn, and winter, represented with grey bars, were 0.11, 014, and 0.21 ACH, respectively, while the mean estimated ventilation rates of summer, autumn, and winter, represented with black bars, were 0.32, 029, and 0.44 ACH, respectively. Recommended ventilation rates for the test room by EN and ASHRAE standards were indicated with dotted red lines. Nevertheless, any of the mean estimates of each season did not satisfy the recommended values. We should open more windows to maintain acceptable or desired indoor air quality.

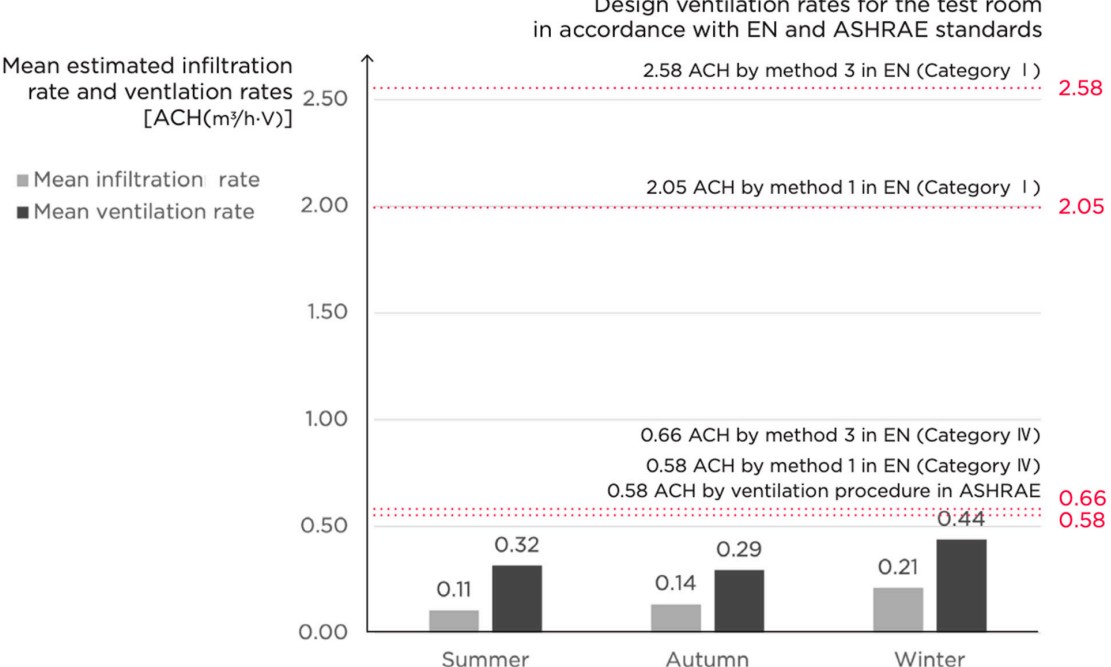

**Figure 15.** Comparison of mean infiltration and ventilation rates of each season and design ventilation rates by EN and ASHRAE standards.

Then, how many circular openings should we open to satisfy the recommended ventilation rates? Suppose that the airflow through the four openings can be calculated by subtracting the infiltration rate from the ventilation rate: ventilation rate (opening scenario 2)—infiltration rate (opening scenario 1). In addition, suppose airflow through

the openings increases at the same rate whenever we open more circular openings on one side and the infiltration of the room is constant. Given those assumptions, we can calculate the expected ventilation rates in the test room when we open more circular openings (infiltration rate) + (airflow only through the four openings = ventilation rate–infiltration rate) × the number of open circular openings/4). Accordingly, we calculated the minimum number of circular openings we should open each season with the mean estimates for each season. In summer, opening ten circular openings on one side can satisfy the minimum ventilation rate (0.64 ACH = 0.11 + (0.32 − 0.11) × 10/4), and in autumn and winter, opening twelve and six circular openings on one side, respectively, can satisfy the minimum ventilation rate (0.59 ACH = 0.14 + (0.29 − 0.14) × 12/4, 0.57 ACH = 0.21 + (0.44 − 0.21) × 6/4). In other words, the test room needs two circular openings per 47 m$^3$, 39 m$^3$, and 78 m$^3$ in summer, autumn, and winter, respectively.

However, these were only rough calculations. The mean estimates of each season cannot represent each season's average infiltration and ventilation rate because the measurement days were irregular and insufficient. In addition, when we open more openings, the ventilation rates might not increase at the same rate due to various factors that could influence the airflow. We should repeat the tracer-gas decay method with the calculated number of circular openings open, and then we can evaluate more accurately whether the minimum ventilation rate is satisfied with those new opening scenarios.

## 5. Conclusions and Significance

This study shows an example of indoor air quality in an office room in Seoul, South Korea. Korea has four distinct seasons with severe temperature changes over a year. Thus, most office buildings operate air-conditioning facilities in summer and winter. However, in many office rooms, workers do not ventilate the room adequately to save energy consumption or because they are indifferent to indoor air quality. In this study, measuring indoor air-quality indicators while all the windows were closed allowed us to see how much indoor air quality could deteriorate if ventilation was not properly performed in actual living conditions. The degree of deterioration may vary in other office rooms depending on their environmental conditions, such as insulating conditions, ventilating conditions, occupants' habits, and heating and cooling system types. Therefore, if one wants to know the indoor air quality in another room, measuring the indoor air quality indicators in the room would be necessary.

We looked into how outdoor temperatures, wind speeds, and $CO_2$ concentrations have changed from summer to winter in Seoul, South Korea. The outdoor temperature in Seoul experiences drastic variation, soaring as high as 35 °C in summer and dropping almost as low as −20 °C in winter. The wind speed in Seoul experiences mild seasonal variation over the course of the year. The mean outdoor $CO_2$ concentration from August to December measured at Namsan Tower was 430 ppm, with 428 ppm in August and 442 ppm in December.

We also looked into how much the outdoor and indoor temperatures differ by season in the test room in an old building when using cooling or heating devices as usual. Then, we estimated how much infiltration occurred when we closed all the windows and how much natural ventilation, including infiltration, occurred when we opened the four circular openings on one side. Further, we analyzed how those estimates changed with varying indoor and outdoor conditions. During the measurement days, the temperature differences in the summer were slight, with an average of 3.6 °C, while the wind speed averaged 2.3 m/s, and the mean estimated infiltration and ventilation rates of summer were 0.11 ACH and 0.32 ACH, respectively. On the other hand, in autumn, the temperature differences increased to an average of 7.8 °C, while the wind speeds were calmer on most days, averaging 1.6 m/s. The mean estimated infiltration and ventilation rates of autumn were 0.14 ACG (127% of summer mean infiltration rate) and 0.29 ACH (0.91% of summer mean ventilation rate), respectively. The temperature differences in winter increased significantly to an average of 15.9 °C. On top of that, the wind speeds increased to an average of 3 m/s

with a wide variance. As a result, the estimated ventilation rates in winter increased significantly compared to the ones in summer and autumn. The mean estimated infiltration and ventilation rates of autumn were 0.21 ACH (191% of summer mean infiltration rate) and 0.44 ACH (138% of summer mean ventilation rate), respectively. The exchanged air due to leaks and airflow through openings increased significantly in winter than in other seasons in Korea. It is because both the temperature difference and the wind speed increased significantly in winter. Therefore, using the windows the same way over a year would not be an optimal ventilation strategy in Korea with four distinct seasons.

Existing buildings in Korea have various types of windows but it is not easy to know how much ventilation occurs with the windows they have. Especially, many old buildings do not have mechanical ventilating facilities installed, so natural ventilation is often the only available ventilation method. In addition, there is a high possibility that a lot of air leakage has occurred in old buildings through nonairtight envelopes. The air exchange caused by the leaks should not be ignored to reduce excessive energy loss. However, because of its complexity and cost, only a few people conduct ventilation measurements to determine the actual magnitude of ventilation through windows and leaks in their buildings. Furthermore, Korea does not have regulations or guidelines that propose the recommended ventilation rates and practicable suggestions to maintain acceptable indoor air quality for various small buildings. The tracer-gas method with occupant-generated $CO_2$ in this study can be done only with $CO_2$ sensors and a few people in any living room. Then, the occupants can evaluate whether they have ventilated the room properly, too little, or too much. If the ventilation has been insufficient, the occupants will be able to pay more attention to improving the indoor air quality by ventilating more. Conversely, they can reduce energy loss by reducing ventilation if it has been excessive. Altogether, the method allows people to easily set up specific ventilation strategies for which windows and for how long they should be open in their room to achieve the desired indoor air quality.

Small circular openings in the test room are not typical window types. A full-scale experiment found that they could achieve sufficient ventilation to maintain acceptable air quality by keeping them open all the time in the test room. Generally, it is recommended to ventilate a room once every few hours with large openings to save energy use in cooling and heating seasons. However, many workers often leave windows closed throughout work hours. It may be because occupants can directly feel the discomfort of temperature fluctuation and city noise when they open the windows. Yet, awareness about the impact of poor indoor air quality is low and opening and closing windows often is a hassle. The small circular openings can maintain the indoor condition that does not fluctuate greatly all the time, leaving only the minimum number of openings open that are adequate for maintaining acceptable air quality by season. It can also prevent excessive heat loss by ventilation. In addition, since there is no risk of a break-in through the small openings, they can be left open even when emptying the room for minimum ventilation during unoccupied periods.

**Author Contributions:** N.K. conceived the original idea; H.S. performed the measurements; N.K., A.B. and D.A. were involved in planning and supervised the work; H.S. processed the experimental data, performed the analysis, drafted the manuscript, and designed the figures; N.K. and A.B. aided in interpreting the results and worked on the manuscript. All authors have read and agreed to the published version of the manuscript.

**Funding:** 1500 dollars were funded by UBLO Inc., Seoul 03056, Republic of Korea.

**Informed Consent Statement:** Informed consent was obtained from all subjects involved in the study.

**Acknowledgments:** Hugo van Santen for the technical support related to air-quality monitors and sensors, Nari Yoon and Robert van Santen for the review on planning and presenting the work.

**Conflicts of Interest:** The authors declare no conflict of interest.

## Appendix A

| No. | Date | Opening Scenario | Initial Indoor $CO_2$ | Indoor $CO_2$ after 1 h | Indoor $CO_2$ after 2 h | Indoor $CO_2$ after 3 h | Daily Mean Outdoor $CO_2$ |
|---|---|---|---|---|---|---|---|
| 1 | 1 August 2022 | 1 | 1492 | 1308 | 1161 | 1076 | 425 |
| 2 | 3 August 2022 | 2 | 1653 | 1247 | 1025 | 878 | 431 |
| 3 | 4 August 2022 | 1 | 1369 | 1284 | 1220 | 1169 | 433 |
| 4 | 24 August 2022 | 2 | 1448 | 1242 | 1069 | 930 | 417 |
| 5 | 25 August 2022 | 1 | 1324 | 1251 | 1174 | 1101 | 418 |
| 6 | 30 August 2022 | 2 | 1476 | 1165 | 928 | 782 | 423 |
| 7 | 31 August 2022 | 1 | 1402 | 1317 | 1241 | 1160 | 425 |
| 8 | 5 October 2022 | 1 | 1281 | 1182 | 1080 | 990 | 439 |
| 9 | 6 October 2022 | 2 | 1385 | 1167 | 1012 | 878 | 438 |
| 10 | 12 October 2022 | 2 | 1544 | 1271 | 1085 | 955 | 442 |
| 11 | 13 October 2022 | 1 | 1223 | 1142 | 1070 | 1004 | 442 |
| 12 | 24 October 2022 | 2 | 1349 | 1068 | 946 | 841 | 439 |
| 13 | 26 October 2022 | 2 | 1538 | 1283 | 1091 | 930 | 448 |
| 14 | 10 November 2022 | 2 | 2368 | 1873 | 1567 | 1334 | 452 |
| 15 | 15 November 2022 | 2 | 1588 | 1239 | 1031 | 879 | 452 |
| 16 | 16 November 2022 | 2 | 1753 | 1355 | 1092 | 907 | 452 |
| 17 | 17 November 2022 | 1 | 1865 | 1657 | 1478 | 1331 | 452 |
| 18 | 20 November 2022 | 2 | 1791 | 1481 | 1266 | 1089 | 452 |
| 19 | 22 November 2022 | 2 | 1479 | 1229 | 1062 | 943 | 452 |
| 20 | 23 November 2022 | 2 | 1510 | 1173 | 976 | 827 | 452 |
| 21 | 24 November 2022 | 2 | 1504 | 1155 | 966 | 835 | 452 |
| 22 | 29 November 2022 | 2 | 1586 | 1055 | 763 | 598 | 452 |
| 23 | 8 December 2022 | 2 | 2229 | 1546 | 1192 | 997 | 448 |
| 24 | 13 December 2022 | 2 | 2592 | 1618 | 1065 | 725 | 432 |
| 25 | 15 December 2022 | 1 | 2089 | 1732 | 1483 | 1283 | 432 |
| 26 | 20 December 2022 | 2 | 1870 | 1402 | 1132 | 938 | 432 |

## Appendix B

| No. | Season | Date | Daily Mean Temperature | Daily Mean Wind Speed | Estimation Period | Number of Valid Estimations | Average Temperature Difference during Time Ranges of Estimates | Average Wind Speed during Time Ranges of Estimates | Average Infiltration Rate or Ventilation Rates of Valid Estimates |
|---|---|---|---|---|---|---|---|---|---|
| 1 | Summer | 1 August 2022 | 28.4 | 2.4 | 9:00~22:40 | 15 | 2.4 | 3.1 | 0.16 |
| 2 | | 3 August 2022 | 27.1 | 2.8 | 16:10~18:00 | 6 | −2.5 | 3.1 | 0.36 |
| 3 | | 4 August 2022 | 28.9 | 2.5 | 18:30~02:30 | 25 | 0 | 2.2 | 0.07 |
| 4 | | 24 August 2022 | 24.7 | 2.5 | 18:40~21:00 | 3 | 2.8 | 2.3 | 0.22 |
| 5 | | 25 August 2022 | 22.3 | 1.8 | 22:00~23:50 | 2 | 5.7 | 1.1 | 0.1 |
| 6 | | 30 August 2022 | 19.1 | 2.6 | 19:00~20:30 | 4 | 6.5 | 2.9 | 0.37 |
| 7 | | 31 August 2022 | 21.6 | 1.8 | 20:20~01:50 | 22 | 5.4 | 1.4 | 0.1 |
| 8 | Autumn | 5 October 2022 | 16.6 | 2.1 | 19:50~22:40 | 12 | 8.2 | 1.8 | 0.14 |
| 9 | | 6 October 2022 | 15.7 | 2.1 | 18:50~20:50 | 5 | 7.6 | 1.8 | 0.27 |
| 10 | | 12 October 2022 | 13.4 | 1.8 | 18:30~21:00 | 5 | 5.2 | 1.1 | 0.25 |
| 11 | | 13 October 2022 | 16.2 | 1.5 | 18:40~21:40 | 4 | 5.4 | 1 | 0.11 |
| 12 | | 24 October 2022 | 10.8 | 2.8 | 19:00~20:20 | 2 | 6.1 | 2.4 | 0.35 |
| 13 | | 26 October 2022 | 12.6 | 1.9 | 18:40~21:10 | 5 | 7.2 | 2 | 0.25 |
| 14 | | 10 November 2022 | 13.5 | 1.3 | 19:00~23:50 | 23 | 6.2 | 1.3 | 0.24 |
| 15 | | 15 November 2022 | 8 | 2 | 19:50~21:50 | 6 | 11.4 | 2.3 | 0.34 |
| 16 | | 16 November 2022 | 8.6 | 1.7 | 19:10~21:30 | 9 | 11.3 | 0.9 | 0.35 |
| 17 | | 17 November 2022 | 10.2 | 1.6 | 19:10~01:40 | 28 | 9.2 | 1.2 | 0.15 |
| 18 | | 20 November 2022 | 13.8 | 1.7 | 17:40~21:20 | 9 | 3.4 | 1.4 | 0.23 |
| 19 | | 22 November 2022 | 10.7 | 1.9 | 19:50~22:10 | 3 | 7.3 | 1.5 | 0.25 |
| 20 | | 23 November 2022 | 11.1 | 2.1 | 19:10~20:40 | 4 | 9.9 | 2 | 0.38 |
| 21 | | 24 November 2022 | 9.3 | 1.5 | 20:30~22:00 | 3 | 10.1 | 1.3 | 0.32 |
| 22 | | 29 November 2022 | 7.9 | 3.5 | 20:20~21:20 | 1 | 18.5 | 3.7 | 0.24 |
| 23 | Winter | 8 December 2022 | 3.9 | 1.4 | 19:20~22:10 | 3 | 13.1 | 0.5 | 0.4 |
| 24 | | 13 December 2022 | 0.6 | 3.5 | 18:20~20:20 | 7 | 16.4 | 4.6 | 0.61 |
| 25 | | 15 December 2022 | −4.6 | 2.5 | 19:20~23:50 | 21 | 17.8 | 2.7 | 0.21 |
| 26 | | 20 December 2022 | −3.7 | 2.2 | 19:40~22:10 | 10 | 13.6 | 3.3 | 0.34 |

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
