# Peer review of "Estimation of Natural Ventilation Rates in an Office Room with 145 mm-Diameter Circular Openings Using the Occupant-Generated Tracer-Gas Method"

_sustainability, doi:10.3390/su15139892_

Round 1

Reviewer 1 Report

Similarity in this paper is higher than the standard of the Journal. It is observed that entire abstract is copied from single source. Also, complete sentences are copied from single source.

More comments:

·       Title of the article should be more concise and brief

·       Abstract Section: The problem statement and research gap are not properly identified. The conclusive remarks are very generic that in absence of wind, temperature difference play significant role and vice versa. Should include some quantitative results.

·       Number of grammatical and typo mistake. Revise paper from a native English speaker or Proof editing services.

·       Improve the introduction section by including the latest research carried out in last 3-5years.

·       The authors should increase their discussion on previous related research and highlight how their study is providing a different approach or adding significantly to what has been done. The authors have to explain what is the new here in comparison with the previous studies. The novelty of the current work should be highlighted in the introduction. Please try to mention a problem that needs solving - in other words, the research question underlying your study clearer

·       Also define and properly mention the research gap in last paragraph of the introduction section.

·       In the conclusion section, instead of values, it is better to compare the result in percentage.

Author Response

First of all, I really appreciate your feedback.

Reviewer: Similarity in this paper is higher than the standard of the Journal. It is observed that entire abstract is copied from single source. Also, complete sentences are copied from single source. 

Author: Could you tell me which sentences are the same with another source? I did write the abstract by myself. I even actually wrote the abstract in Korean first and translated it and revised, revised, and revised it. I really didn’t copy it. They could be the same accidentally. I’ll rephrase them. I really don’t understand it..

More comments:

R:Title of the article should be more concise and brief

A: changed briefly

R: Abstract Section: The problem statement and research gap are not properly identified. The conclusive remarks are very generic that in absence of wind, temperature difference play significant role and vice versa. Should include some quantitative results.

A: identified problem statement and research gap in line 18-22, and included some quantitative results in line 34, 36

R: Number of grammatical and typo mistake. Revise paper from a native English speaker or Proof editing services.

A: Could you point them out?

R: Improve the introduction section by including the latest research carried out in last 3-5years. 

A: Included in line 62, line 103~119,

R: The authors should increase their discussion on previous related research and highlight how their study is providing a different approach or adding significantly to what has been done. The authors have to explain what is the new here in comparison with the previous studies. The novelty of the current work should be highlighted in the introduction. Please try to mention a problem that needs solving - in other words, the research question underlying your study clearer. Also define and properly mention the research gap in last paragraph of the introduction section.

A: Included previous related research in line 103~119, in line142~148, mentioned the novelty of the current work in line 160~162

R: In the conclusion section, instead of values, it is better to compare the result in percentage. 

A: Included line 739~751

Reviewer 2 Report

The presented paper is structure very well. The experiment and results obtain from that are very well presented. I have one comment please subscrit all the units for example table 1 3000m2. Also may be if the table 4 to be on 1 page.

Author Response

First of all, I really appreciate your feedback.

Reviewer: The presented paper is structure very well. The experiment and results obtain from that are very well presented. I have one comment please subscrit all the units for example table 1 3000m2. Also may be if the table 4 to be on 1 page.

Author: revised Table 1 and Table 4

Reviewer 3 Report

The manuscript entitled “Estimation of infiltration and ventilation rates in an office room with 145mm-diameter circular openings using the occupant-generated carbon dioxide tracer gas method” is interesting. However, there are some concerns shall be addressed by the authors. Also, the manuscript shall be sent for proofread. Noticeable grammatical errors were identified throughout the manuscript.

1.       Abstract- line 31- “wind speeds were calm” is subjective. Kindly provide the range of air velocity that authors considered it as stagnant/ calm.

2.       Abstract- line 32- “temperature difference were low” is also subjective. Authors are required to provide the range to claim that the difference is low.

3.       Introduction- Ventilation rate is one of the major factors in the present study. However, the ventilation rate (air change rate) is not comprehensive in the discussion section. Kindly include more information (similar/ contradict findings reported in past studies). It is important to link it to the gap (present study) that trying to fill in.
https://doi.org/10.3390/buildings13020459
https://doi.org/10.1007/s10973-022-11466-6

4.       Section 2.1- Figure 1 (b)- Major dimension, labelling (inlet and outlet with details, such as velocity & turbulent intensity shall be included). Since the manuscript mentioned north, west, south walls, kindly label such info in the figure.

5.       Line 168- all the mentioned furniture shall be indicated in figure.

6.       Table 2 and table 3 can be merged in to in table.

7.       Result and discussion are comprehensive.

8.       Line 52- Reference is needed to support the statement. Also, do revise the sentence to enhance readability.

9.       Line 49 – “Contaminants from the occupants (bio effluents) can deteriorate people’s health and well-being”. Reference is needed. A recent manuscript mentioned this statement.
 https://doi.org/10.1016/j.buildenv.2023.110048

10.   Table 1- Superscript for m2 shall be revised.

11.   Figure 3(a – d) need to include more labelling to clearly deliver the idea to the readers.

12.   Abbreviation of PM shall be written in full when first time introduce it. Why PM was measured in the study, but does not reflect in the title/ objective of the study? Kindly provide justification/ rebuttal.

Shall sent for proofread.

Author Response

First of all, I really appreciate your kind feedback :)!

Reviewer: The manuscript entitled “Estimation of infiltration and ventilation rates in an office room with 145mm-diameter circular openings using the occupant-generated carbon dioxide tracer gas method” is interesting. However, there are some concerns shall be addressed by the authors. Also, the manuscript shall be sent for proofread. Noticeable grammatical errors were identified throughout the manuscript.

Reviewer: Abstract- line 31- “wind speeds were calm” is subjective. Kindly provide the range of air velocity that authors considered it as stagnant/ calm.

Author: included some quantitative results in line 34, 36

R: Abstract- line 32- “temperature difference were low” is also subjective. Authors are required to provide the range to claim that the difference is low.

A: included some quantitative results in line 34, 36

R: Introduction- Ventilation rate is one of the major factors in the present study. However, the ventilation rate (air change rate) is not comprehensive in the discussion section. Kindly include more information (similar/ contradict findings reported in past studies). It is important to link it to the gap (present study) that trying to fill in.
https://doi.org/10.3390/buildings13020459
https://doi.org/10.1007/s10973-022-11466-6

A: Included previous related research in line 103~119, in line142~148, mentioned the novelty of the current work in line 160~162

R: Section 2.1- Figure 1 (b)- Major dimension, labelling (inlet and outlet with details, such as velocity & turbulent intensity shall be included). Since the manuscript mentioned north, west, south walls, kindly label such info in the figure.

A: Revised Figures 1 (b)

R:Line 168- all the mentioned furniture shall be indicated in figure.

A: Revised Figures 1 (a)

R: Table 2 and table 3 can be merged in to in table. 

A: I think they would be better to be separated because table 4 is also another method (method 3) suggested in the same EN standard like method 1 in table 2  and method 2 in table 3. If I merge table 2, 3, 4 altogether in one table, it would be too much. But If I only merge two among three, it can be confusing.

R: Result and discussion are comprehensive.

A: explained with quantitative results line 727~751

R: Line 52- Reference is needed to support the statement. Also, do revise the sentence to enhance readability.

A: Revised in line 65

R: Line 49 – “Contaminants from the occupants (bio effluents) can deteriorate people’s health and well-being”. Reference is needed. A recent manuscript mentioned this statement. 
 https://doi.org/10.1016/j.buildenv.2023.110048

A: Revised in line 61

R:Table 1- Superscript for m2 shall be revised.

A: Revised Table 1

R: Figure 3(a – d) need to include more labelling to clearly deliver the idea to the readers.

A: Revised Figure 3

R: Abbreviation of PM shall be written in full when first time introduce it. Why PM was measured in the study, but does not reflect in the title/ objective of the study? Kindly provide justification/ rebuttal.

A: It is written in line 193. PM is Particulate Matter. There was an typo about it..To evaluate indoor air quality, PM concentration (fine dust concentration) is also an important factor. Fine dust in the air has a detrimental effect on human health. In Korea, there are many days with severe fine dust pollution, so it is an issue of high interest. Bio-effluent can be expelled through natural ventilation, but PM rather comes in from the outside when natural ventilation is used. In Korea, it is recommended to close the windows indoors on days when the PM concentration is high. Therefore, before opening the window for natural ventilation, the external PM concentration should also be considered, so it was measured. During the days observed in this study, there were no days where the external PM concentration was harmfully high, so ventilation was more advantageous to IAQ. These contents were briefly mentioned in line 525 and were not emphasized in the title or objective because they were not that important in this study.

Round 2

Reviewer 1 Report

Author addressed all the comments and article is now suitable for publication.

Should proofread the article for minor grammatical / typo corrections.

Author Response

Thanks for the feedback!

I corrected typo and some sentences.

Reviewer 3 Report

The technical part shall be heavily revised. Some information sounds inappropriate.

Discussion and conclusion make sense.

English is ok.

Author Response

Thanks for the feedback!

Reviewer: The technical part shall be heavily revised. Some information sounds inappropriate.

Author: I revised section 1 and 2.

Round 3

Reviewer 3 Report

Generally, the presented results/ works are sufficient. Figures have been improved accordingly. However, the write-up especially the discussion and method shall be heavily revised.

1) Presenting the data with brief descriptions is not sufficient. Do compare your finding with previous studies, and provide some justification for the contradicting findings.

2) What is the main information/ idea that authors would like to highlight to the readers in Fig 14? As of current form, readers know the range of temperature differences and ventilation rates only.

Author Response

R: However, the write-up especially the discussion and method shall be heavily revised.

A: 4. discussion and method part and 2.4 Measurement set-up and calculations  are revised.

R: Presenting the data with brief descriptions is not sufficient. Do compare your finding with previous studies, and provide some justification for the contradicting findings.

A: changed some expressions and revised figures in 3. Result and 2.4 Measurement set-up and calculations  are revised.

2) What is the main information/ idea that authors would like to highlight to the readers in Fig 14? As of current form, readers know the range of temperature differences and ventilation rates only.

A: Figure 14 shows various environmental conditions in different seasons of South Korea of the test room and, subsequently, various infiltration and ventilation rates estimated in each season. (line 574). 

As a result, the estimated ventilation rates in winter increased significantly compared to summer and autumn even though the same openings were open. (line 611)
